# Dynamics of collagen oxidation and cross linking in regenerating and irreversibly infarcted myocardium

Eman A. Akam-Baxter [1,2] ✉, David Bergemann[1], Sterling J. Ridley[1], Samantha To[1], Brittany Andrea[1], Brianna Moon [3], Hua Ma[3], Yirong Zhou[1,4], Aaron Aguirre [1,4], Peter Caravan[2,3], Juan Manuel Gonzalez-Rosa [1,5] & David E. Sosnovik[1,2,3]

In mammalian hearts myocardial infarction produces a permanent collagen-rich scar. Conversely, in zebrafish a collagen-rich scar forms but is completely resorbed as the myocardium regenerates. The formation of cross-links in collagen hinders its degradation but cross-linking has not been well characterized in zebrafish hearts. Here, a library of fluorescent probes to quantify collagen oxidation, the first step in collagen cross-link (CCL) formation, was developed. Myocardial injury in mice or zebrafish resulted in similar dynamics of collagen oxidation in the myocardium in the first month after injury. However, during this time, mature CCLs such as pyridinoline and deoxypyridinoline developed in the murine infarcts but not in the zebrafish hearts. High levels of newly oxidized collagen were still seen in murine scars with mature CCLs. These data suggest that fibrogenesis remains dynamic, even in mature scars, and that the absence of mature CCLs in zebrafish hearts may facilitate their ability to regenerate.

In adult mammals myocardial infarction (MI) results in the formation of a permanent collagen-rich scar in the injured myocardium[1]. While the scar provides tensile strength to the infarct and prevents its rupture, it also creates the substrate for adverse electrical and mechanical remodeling of the heart[2]. In zebrafish the initial response to an acute cardiac injury is similar and also results in the formation of a collagen-rich scar[3–5]. However, unlike the mammalian heart, the fibrotic scar in zebrafish hearts is completely resorbed during the regeneration process[3,6].

The collagen molecules forming a scar are initially deposited as single fibers, which form a triple helix. As the scar matures covalent cross-links form between the individual collagen fibers, and the extent of these is thought to modulate the ability of collagen to be degraded[7,8]. Collagen cross-link (CCL) formation is initiated by enzymatic, lysyl oxidase-mediated, oxidation of lysine and hydroxylysine residues to form the lysine aldehydes (Lys[Ald]) allysine and hydroxy-allysine. These aldehydes spontaneously react with other collagen Lys[Ald] or with amino groups on collagen lysine residues. The resulting aldehyde cross-links undergo further rearrangements and reactions to produce degradation-resistant CCLs such as pyridinoline and deoxypyridinoline[9]. Differences in the extent and nature of collagen cross-linking between species could play an important role in determining whether scarred myocardium can be regenerated. However, CCL formation has not been previously characterized and compared in zebrafish and murine hearts.

The initial and crucial step in the CCL cascade is the formation of Lys[Ald] [10]. We have previously developed gadolinium-containing molecular imaging probes that can bind to Lys[Ald] and quantify collagen cross-

[1]Cardiovascular Research Center, Cardiology Division, Massachusetts General Hospital, Harvard Medical School, Boston, MA, USA. [2]Institute for Innovation in Imaging, Massachusetts General Hospital, Boston, MA, USA. [3]A.A. Martinos Center for Biomedical Imaging, Massachusetts General Hospital, Harvard Medical School, Boston, MA, USA. [4]Wellman Center for Photomedicine, Massachusetts General Hospital, Harvard Medical School, Boston, MA, USA. [5]Biology Department, Boston College, Chestnut Hill, USA. ✉e-mail: eakam@mgh.harvard.edu

linking activity (fibrogenesis) by MRI[11,12]. These probes have been validated in several models of fibrosis, including pulmonary, hepatic, and renal fibrosis[13], but are not suited to imaging the small zebrafish heart. Additionally, biochemical methods to asses Lys[Ald] require relatively large amounts of tissue[14]. Consequently, to enable the study of CCLs in zebrafish hearts, we developed a library of fluorescent Lys[Ald] binding probes. All of the probes contain an aldehyde-binding motif linked to 6-carboxy-tetramethyl rhodamine (TAMRA) as a fluorescent reporter. The binding and reactivity of these compounds was tested in vitro and their performance in vivo was tested using the thoracic aorta, which contains large amounts of Lys[Ald], as a positive control. The oxyamine-bearing probe TMR-O emerged as the best probe and was used to characterize the natural history of collagen oxidation and cross-linking in zebrafish and murine models of myocardial infarction/injury. In addition, high performance liquid chromatography was used to characterize the levels of mature, degradation-resistant cross-links in the murine and zebrafish hearts. Our results indicate that, while the initial oxidation and cross-linking of collagen is similar in murine and zebrafish hearts, significant differences in the hydroxylation of lysine and formation of mature degradation-resistant cross-links exist between the two species. These differences in CCLs have not been previously described and provide additional insights into the regenerative competence of the zebrafish compared to the mammalian heart.

## Results

### Synthesis and reactivity of fluorescent lysine-aldehyde binding probes

Gadolinium-based probes are frequently designed to have rapid binding and dissociation kinetics, which facilitates their clinical translation[15]. Here, however, we aimed to synthesize a library of fluorescent Lys[Ald]-binding probes with a range of reaction kinetics and reactive moieties. The sensitivity of fluorescent imaging is a major advantage for targeted imaging in the zebrafish heart, which presents very low levels of target due to its extremely small size. Synthesis of a library of fluorescent probes allowed us to select a conjugate that would bind quickly and reversibly to tissue aldehydes, but also show sufficiently slow reversibility to allow for fluorescent microscopy of tissue sections and even serial imaging. The Lys[Ald]-reactive fluorescent probes in this work feature TAMRA as the fluorescent reporter. The aldehyde-reactive group was attached to TAMRA via an amide linkage to the 6-carboxy position, which preserves the fluorescence.

A number of aldehyde-binding groups with varying structures (Fig. 1a) and binding mechanisms (Fig. 1b) were synthesized. The TMR-O probe bears an alkyl oxyamine, while TMR-LHZ and TMR-HZN feature hydrazines. However, unlike the commercially available TAMRA hydrazide (TMR-HZD, Lumiprobe), TMR-HZN features a secondary piperazinyl alkyl hydrazine group that has superior reactivity to the hydrazide binding group[11]. We additionally found that commercial TMR-HZD is insoluble in purely aqueous buffer, while TMR-HZN is readily soluble in aqueous solutions. TMR-NB is the non-binding (non-reactive) analog of TMR-HZN, featuring TAMRA and the cyclic piperazine group but lacking the reactive hydrazine. TMR-Pyr and TMR-Rho react with aldehydes via non-reversible iso-Pictet Spengler and Knoevenagel linkages respectively. While the iso-Pictet Spengler involves a reversible condensation step[16], the condensation reaction is followed by a cyclization step, allowing TMR-Pyr to irreversibly react with aldehydes. For TMR-Rho, the Knoevenagel ligation of the heterocyclic rhodanine moiety is reported to be irreversible, resulting in an aldol-like condensation product[17].

The reactivity of the synthesized probes towards aldehydes was tested in vitro using butyraldehyde as a model alkyl aldehyde with similar structure to the in vivo aldehyde target allysine (Supplementary Fig. S1). We used pseudo-first order conditions in which the aldehyde was at 10-fold higher concentration than the reactive probe.

Formation of the butyraldehyde adduct was monitored by HPLC with fluorescence detection (Fig. 1c–h) and the observed rate constants ($k_{obs}$) were measured for each reaction (Eq. 3, methods). The table in Fig. 1b provides a summary of the measured rate constants for the probe reactions with butyraldehyde at physiological pH and ambient temperature. As expected, we found that the non-binding control probe TMR-NB showed no reactivity with butyraldehyde. Of the series of reactive probes, the oxyamine probe TMR-O had the fastest on-rate for the reaction, while the commercially available TAMRA hydrazide (TMR-HZD) was the slowest. The reaction rate of TMR-O was 27-fold higher than TMR-HZD. The remarkable difference in reactivity between probes is also evident from the HPLC traces for the reaction of TMR-O, TMR-HZN and TMR-HZD (Fig. 1c, e, and g) where at 56 min after the start of the reaction almost complete conversion is observed for TMR-O, less than 50% conversion for TMR-HZN, and minimal conversion for TMR-HZD. Additionally, for the commercial TMR-HZD, we found a small but significant impurity (Fig. 1c, d, peak shown in black), which we could not identify by HPLC-MS. Notably, for the rhodanine-containing TMR-Rho, we observed multiple reaction products with butyraldehyde, most of which we were not able to identify by HPLC-MS (Supplementary Fig. S2a). Because of the complexity of the HPLC traces for the reaction of TMR-Rho with butyraldehyde, we were unable to calculate a meaningful rate constant. Additionally, we found that TMR-Rho is non-selective, and reacts avidly with amines as well as aldehydes (Supplementary Fig. S2b).

The oxime product of TMR-O is remarkably stable to hydrolysis, and no reaction reversibility was observed in the first hour of the reaction (Fig. 1h), and even after 24 h (Supplementary Fig. S3). For the hydrazones, the TMR-HZD adduct is much more stable to hydrolysis than the TMR-HZN adduct (Fig. 1d, and f), which is likely due a stabilization effect from the acyl group of TMR-HZD[18]. Since oxyamines are reported to react slower than hydrazines at physiological pH[19], we further investigated the reason for the very high reaction rate observed for TMR-O in this study.

We hypothesized that the cyclic, secondary hydrazine in TMR-HZN provides steric hindrance that slows the rate of hydrazone formation relative to a linear alkyl moiety such as that in TMR-O. We therefor synthesized a linear hydrazine analog, TMR-LHZ which features an almost identical structure to TMR-O, but with a hydrazine reactive group (Fig. 1a). The reaction rates for TMR-O and TMR-LHZ are near identical, confirming the faster reactivity of the linear alkyl vs. cyclic hydrazine (i.e., TMR-LHZ vs TMR-HZN Fig. 1i). While the on-rates for reaction of TMR-O and TMR-LHZ were similar, marked differences were observed in the extent of product formation. At 80 min after the start of the reaction, TMR-LHZ resulted in less than 50% product formation, while more than 80% of TMR-O was converted to the condensation product (Fig. 1j, k).

### TMR-O shows superior binding to protein aldehydes in vitro and to tissue aldehydes in vivo

The kinetics of fluorescent probes can be altered in biological milieus with large amounts of charged and hydrophobic/hydrophilic moieties. We therefore next tested the binding of the fluorescent library of probes to Lys[Ald] side chains to bovine serum albumin that had been chemically oxidized to convert lysines to allysines (BSA-Ald) (Fig. 2a). Solutions of BSA-Ald (25 μM protein, 100 μM aldehyde) were incubated with 10 μM of TAMRA-based probes for 3 h at 37 °C. UV-Vis absorption spectra were then collected, and the protein-bound probe was separated from unbound probe by ultrafiltration using a 10 kD molecular weight cut-off filter (MWCO). UV-Vis spectra were then collected for the filtrate and compared to the spectrum measured before separation, either with or without blocking (Fig. 2b and Fig. 2c). We found that the extent of binding to protein-aldehydes was highest for TMR-O, followed by TMR-Pyr, and lowest for the non-binding probe TMR-NB. Despite differences in the reaction rates, we found that the extent of

BSA-Ald binding for TMR-HZN, TMR-Rho and TMR-HZD were very similar, about 20% (Fig. 2d). Blocking of the aldehyde sites on BSA-Ald with methoxyamine almost completely eliminated binding of the compounds to BSA-Ald, confirming the specificity of our probes to aldehydes (Fig. 2c–e).

To evaluate the in vivo performance of the probes, we next examined their binding to healthy aldehyde-rich tissues in vivo. Mammalian aorta and zebrafish bulbus are similar in composition and function[20]. Both the aorta and bulbus are rich in elastin, which allows them to withstand dynamic increases in pressure during systole[21,22].

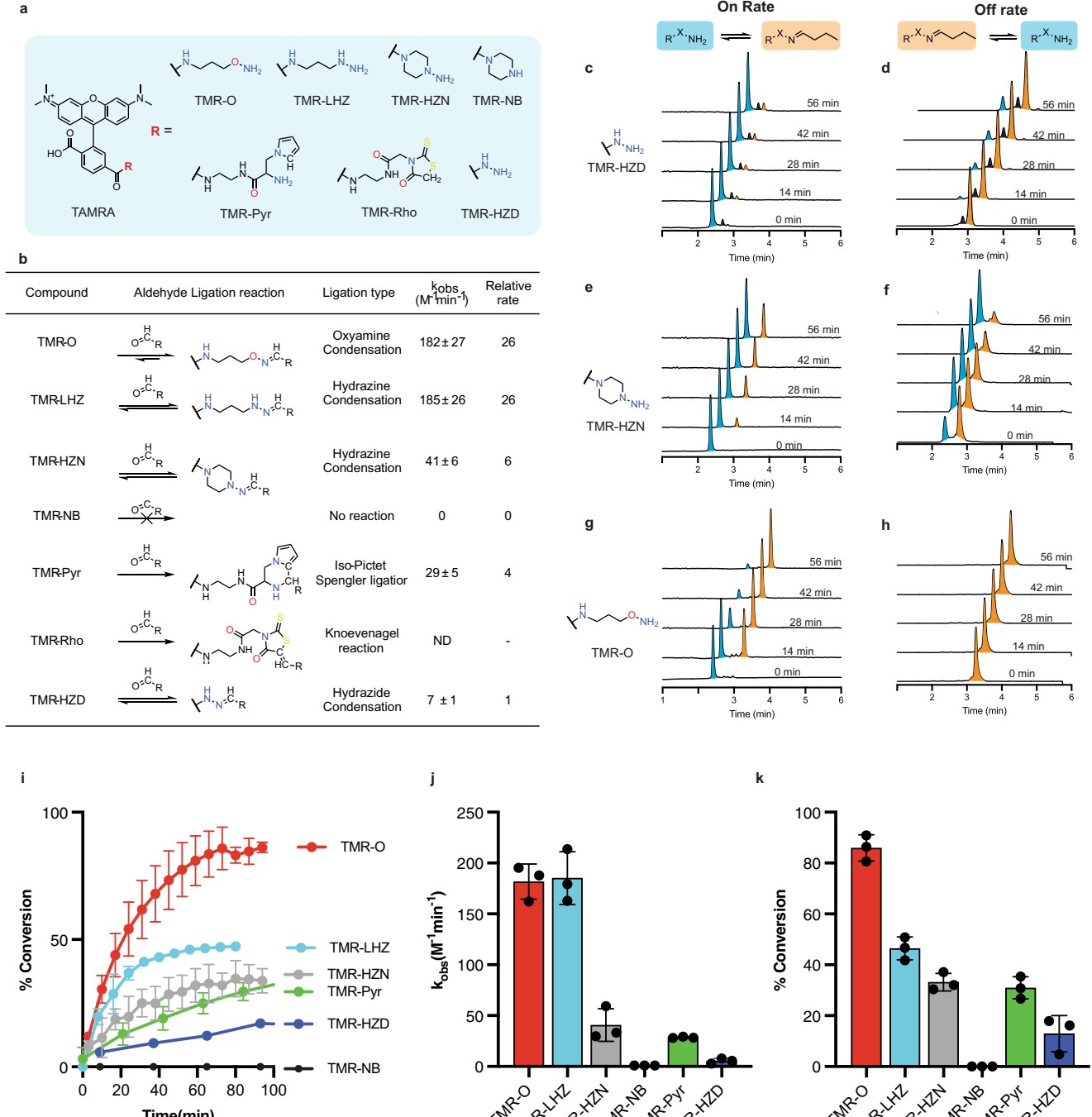

**Fig. 1 | Reactivity of a library of aldehyde-binding fluorescent probes.**
**a** Chemical structure of the synthesized TAMRA-based probes. **b** Aldehyde-ligation reaction types and corresponding reaction rate constants relative to the commercially available TMR-HZD. The rate constant of the oxyamine probe (TMR-O) is 27-fold higher than TMR-HZD. **c**–**h** Reaction rates of the probes with butyraldehyde were followed by HPLC with fluorescence detection. HPLC traces for the forward (**c**, **e**, **g**) and reverse (**d**, **f**, **h**) reactions for TMR-HZD, TMR-HZN and TMR-O with butyraldehyde showing the ligation product (orange) and the free, unbound probe (blue). Of note, the HPLC traces for the commercially available TMR-HZD (**c**, **d**) also contain a small unidentified impurity (black). **i** Plots of the observed rate constant for the formation of the aldehyde-ligation products, (**j**) the corresponding percent conversion as a function of time and, (**k**) the percent product generated at 100 min. The non-binding control probe (TMR-NB), in which a cyclic piperazine group replaces the reactive hydrazine, shows no reactivity, confirming the specificity of the oxyamine (TMR-O), hydrazine (TMR-HZN) and pyrrole (TMR-Pyr) based TAMRA probes for aldehydes. TMR-O, however, has the fastest reactivity and highest percent conversion to ligation product. Values are plotted as the average of three experiments ± standard deviation. Source data are provided in the Source Data file.

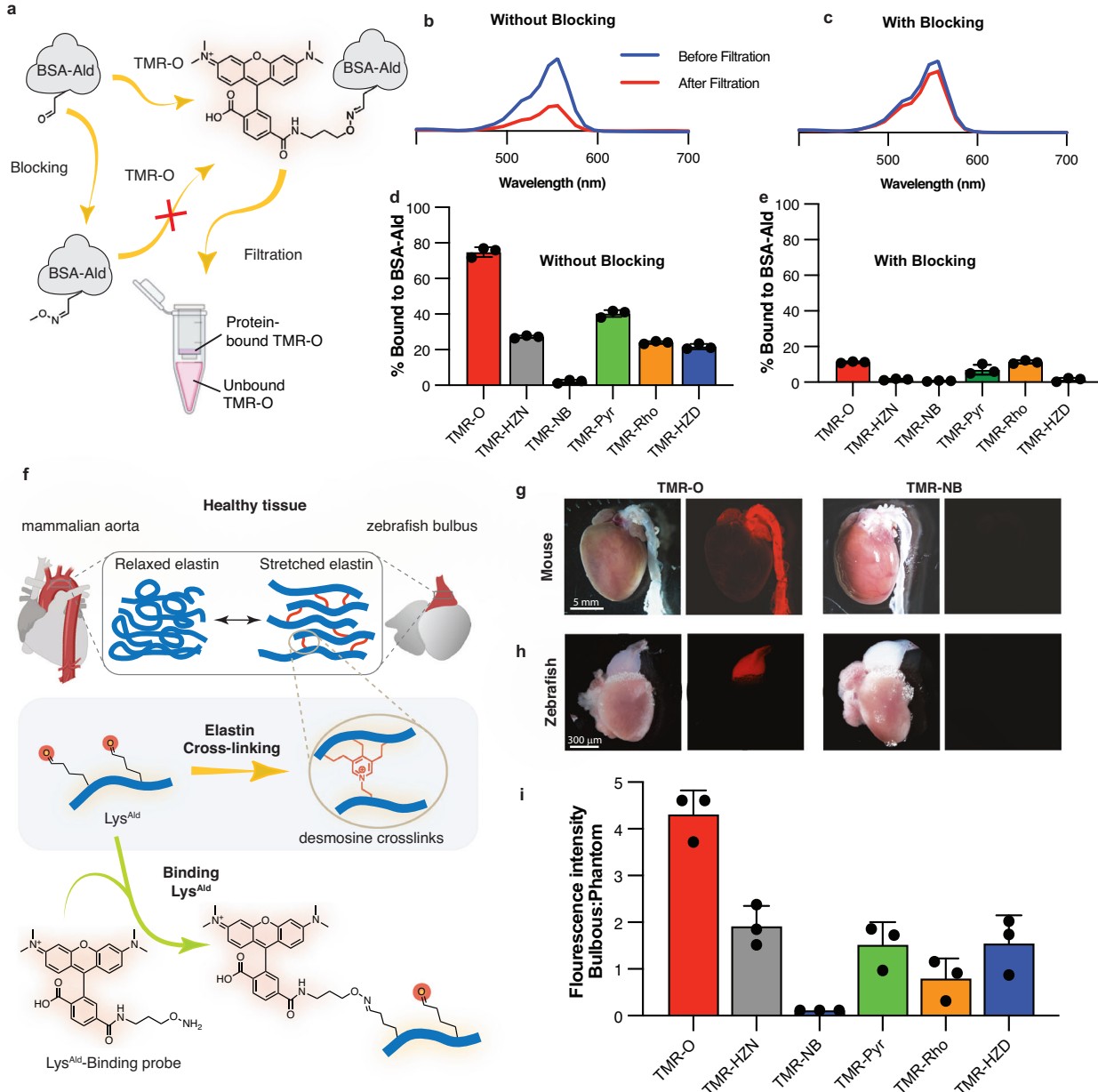

**Fig. 2 | TMR-O shows superior selective binding to protein and tissue aldehydes.** Created in part with Biorender.com. **a** Binding of the probes to aldehyde-rich protein (BSA-Ald) was determined by incubating the probes (10 mM) with BSA-Ald (25 mM protein, 100 mM aldehyde) with and without aldehyde-blocking with methoxyamine. After incubation at 37 °C for 3 h, free and protein-bound probe were separated by passing solutions through a 10 kD molecular weight cut-off (MWCO) filter. **b** UV-Vis spectra of a solution of TMR-O incubated with BSA-Ald before (blue) and after (red) passing through a MWCO filter show a significant decrease in the TMR-O signal due to binding to BSA-Ald. (**c**) Blocking of the aldehyde-reactive sites on BSA-Ald obstructs TMR-O binding, resulting in little change in the UV-Vis absorption of TMR-O after filtration. **d**, **e** Quantification of the percent binding of aldehyde-reactive TAMRA-based probes to BSA-Ald without (**d**) and with (**e**) methoxamine pre-blocking demonstrates their specificity for aldehydes. **f** Lysine aldehydes (Lys[Ald]) are abundant in healthy mammalian aorta and zebrafish bulbus due to elastin turn-over and cross-linking (top). These tissues, therefore, provide a robust control to characterize the accuracy of TAMRA-based aldehyde-reactive probes for lysine aldehydes such as allysine and hydroxy-allysine. **g**, **h** White light and fluorescence reflectance images of whole mouse (**g**) and zebrafish (**h**) hearts excised after systemic injection of TMR-O or TMR-NB. TMR-O binds specifically to the aorta and bulbus, while the non-binding control probe TMR-NB produces no signal. Imaging was performed in 3 mice/fish per group with consistent results. **i** Quantification of fluorescence signal intensity in the zebrafish bulbus 4 h after IP injection of an allysine-binding probe (2 nmol/g) shows > 2-fold higher signal for TMR-O compared to other probes and no uptake of TMR-NB. Signal is normalized to the fluorescence signal of a phantom made from the injected dose of each probe. Data are means ± SD of three independent experiments. Source data are provided in the Source Data file.

The visco-elastic properties of elastin in these tissues are attributable to desmosine cross-links that are initiated by allysine aldehydes (Fig. 2f)[9,23]. Since elastin in the aorta and bulbus is rich in Lys[Ald][14], we hypothesized that our probes would accumulate in both healthy aorta and bulbus.

Intravenous injection of TMR-O (2 nmol/g) in healthy mice resulted in a strong accumulation of the probe in the aorta 90 min after injection (Fig. 2g), while intravenous administration of an equivalent dose of the non-binding probe TMR-NB did not result in any observable fluorescence in the aorta (Fig. 2g). In zebrafish, probes were

injected intraperitonially at the same dose of 2 nmol/g, and the hearts were excised 4 h after injection. Similarly, TMR-O but not TMR-NB showed robust accumulation in healthy zebrafish bulbus (Fig. 2h). To evaluate the performance of probes in vivo, we quantified the accumulation of each probe in the zebrafish bulbus (Fig. 2i, and Supplementary Fig. S4). TMR-O fluorescence was more that two-fold higher than all other compounds.

Despite very similar rates for the forward reactions of TMR-O and TMR-LHZ, the accumulation of TMR-LHZ in zebrafish bulbus was 4-fold lower. Additionally, although TMR-HZD reacted 6-times slower than TMR-HZN and TMR-Pyr, all three compounds accumulated on the bulbus to the same extent (Fig. 2i). As expected, the non-binding control probe TMR-NB did not bind to bulbus. Collectively, these results demonstrate the excellent specificity of our probes to protein and tissue aldehydes and indicate that the hydrolysis/ dissociation kinetics of the aldehyde-bound adducts govern the accumulation of these probes on $Lys^{Ald}$-rich proteins and tissue in vivo.

## Collagen oxidation and cross-linking in the zebrafish model of myocardial cryoinjury

Based on its superior performance, we selected the allysine-binding probe TMR-O to investigate the natural history of collagen oxidation and cross-linking in a zebrafish model of myocardial necrosis. Injury was created using the well-established cryoinjury model[6], where a liquid-nitrogen cooled metal probe is used to injure ~25% of the zebrafish ventricle. This model simulates the acute injury that occurs in myocardial infarction[6]. As collagen is deposited in the healing injury, it is stabilized by cross-linking, which is initiated by lysyl-oxidase-mediated oxidation of lysine residues[8]. The resulting oxidized collagen is rich in $Lys^{Ald}$, which can bind TMR-O (Fig. 3a). Cryoinjuries were performed in transgenic animals expressing nuclear GFP under a cardiomyocyte-specific promoter[24,25], which allowed us to distinguish the injured region based on the absence of GFP expression.

To determine the dynamics of collagen oxidation and cross-linking in the injured zebrafish heart, the fish were injected with TMR-O (2 nmol/g, IP) at multiple time points after injury, euthanized 4 h after injection, following which the whole hearts were imaged ex vivo. Figure 3b shows the transmitted light, green fluorescence, and red fluorescence images of zebrafish hearts at 7, 10-, 15-, 21- and 35 days post injury (dpi). In all cases, the GFP images showed a lack of green fluorescence in the injury-zone and in the bulbus (Fig. 3b). The red fluorescence images of TMR-O showed a consistent signal in the bulbus at all stages after injury (Fig. 3b). In the injury, however, the TMR-O signal was weak at 7 dpi but then increased from days 10–35 post injury. Injection of injured zebrafish with the non-binding control probe Quantification of the fluorescence signal in the zone-of-injury, normalized to the uninjured area of the ventricle, showed a significant increase in fluorescence at 21 and 35 dpi vs. days 7 and 10 dpi (Fig. 3c, $p < 0.001$, one-way ANOVA, comparison between 7 vs. 21, 7 vs. 35 dpi and 10 vs. 21 and 10 vs. 35 dpi). The TMR-O signal in the injured myocardium peaked at 21 dpi and remained high at 35 dpi consistent with ongoing fibrogenesis.

We next sought to correlate the distribution and intensity of the TMR-O signal with the dynamics of scar formation. Following whole-heart imaging, the hearts were sectioned and stained with the Acid Fuchsin-Orange G (AFOG) technique for detection of scar tissue. The slow hydrolysis of the TMR-O oxime (Fig. 1h) allowed fluorescence microscopy of unstained sections to be compared with adjacent AFOG-stained histological sections in which collagen (blue), fibrin (red) and normal muscle (brown) are highlighted (Fig. 3d). In all cases, a strong overlap was observed between collagen-positive areas by histology (blue) and areas of TMR-O fluorescence (Fig. 3e).

Infarct volume was quantified through the absence of GFP signal in the heart by fluorescence microscopy. As described[3], we detected a significant reduction in the infarct volume at 21 and 35 dpi, compared to 7 and 10 dpi (Fig. 3f, $p < 0.006$, one-way ANOVA), consistent with the

initiation of the regenerative process. While the size of the injured zone decreased with time, the relative amount of collagen within this zone, assessed by AFOG staining, increased with time and collagen was apparent starting at 10 dpi (Fig. 3g). TMR-NB did not result in any fluorescence in the bulbus or the area of injury, both on whole-heart imaging and by fluorescence microscopy (Fig. 3h), confirming that the specificity of TMR-O for $Lys^{Ald}$ was retained in vivo.

Collectively these data show that, in the 35 days following acute injury in zebrafish, a portion of the injured myocardium does heal and regenerate, but high levels of collagen deposition and cross-linking persist, suggesting that in the residual injury the process of collagen deposition and resolution remains highly dynamic.

## Collagen cross-linking dynamics in a mouse model of myocardial infarction

We next evaluated the oxidation and cross-linking of collagen in infarcted mammalian hearts, which lack the regenerative capacity of zebrafish. Permanent ligation of the left coronary artery was performed in adult mice, which were then imaged at 5, 10, 21 and 35 days after injury. The infarcted mice were injected with TMR-O (2 nmol/g) via a tail-vein catheter, and their hearts were excised for ex vivo imaging 90 min after injection. Figure 4 shows bright field and fluorescence reflectance images of hearts of infarcted mice. TMR-O uptake was robust in the aorta and did not vary over time (Fig. 4a). Within the injured area, the uptake of TMR-O peaked in mice at 21 dpi, as it did in zebrafish. However, the peak target/background (infarct/remote) values of TMR-O uptake in the mice were substantially higher than in the zebrafish. In addition, the uptake of TMR-O in the mice began earlier and was close to its peak value already at day 10 dpi (Fig. 4a–c, $p < 0.005$, one-way ANOVA with Tukey's post hoc comparison). Short axis slices of the hearts (Fig. 4b) showed that the uptake of TMR-O was uniform throughout the infarct.

Histological analysis was performed on tissue sections at the midventricular level. Staining with AFOG showed minimal collagen deposition at 5 dpi (Fig. 4d), and a corresponding absence of TMR-O signal in confocal microscopy images at 5 dpi (Fig. 4e). At 10–35 dpi, AFOG-stained sections showed that significant portions of the ventricle stain positive for collagen (Fig. 4d), and that these areas correlated well with areas in adjacent sections that were positive for TMR-O (Fig. 4e). Collectively, these data show that TMR-O retains its specificity for oxidized collagen in mice and, notwithstanding an earlier rise and higher peak in mice, that the oxidation and cross-linking of collagen during the first 35 days of injury is similar in mice and zebrafish.

## Collagen deposition and cross-linking persists in mature scar in the mouse model of MI

The fibrotic scar in MI has been historically considered to mature within 3–4 weeks of injury at which time an inert scar is formed and collagen deposition abates[26,27]. However, at both 45 and 62 days after infarction large portions of the murine infarcts still showed high levels of TMR-O uptake (Fig. 5a–d), indicating robust collagen oxidation. The periphery of the remodeling infarcts was most likely to show high levels of TMR-O uptake, and static portions of the infarct with little TMR-O uptake were most likely to be found at the center of the infarct. Nonetheless, significant levels of probe uptake were seen in the center of the infarct as well (Fig. 5a–d). Pixel classification with k-means clustering showed that the size and mean value of the pixel cluster with the highest amount of probe uptake did not change as the scar matured over time (Fig. 5e–g). The heterogeneous nature of TMR-O uptake in the day 45 and 62 infarcts was confirmed by histology and fluorescence microscopy (Fig. 5h, i). Collectively, the data in the day 45–62 murine infarcts reveals that some areas of these mature infarcts are static, with little deposition and cross-linking of new collagen. However, large portions of the infarcts, particularly towards the

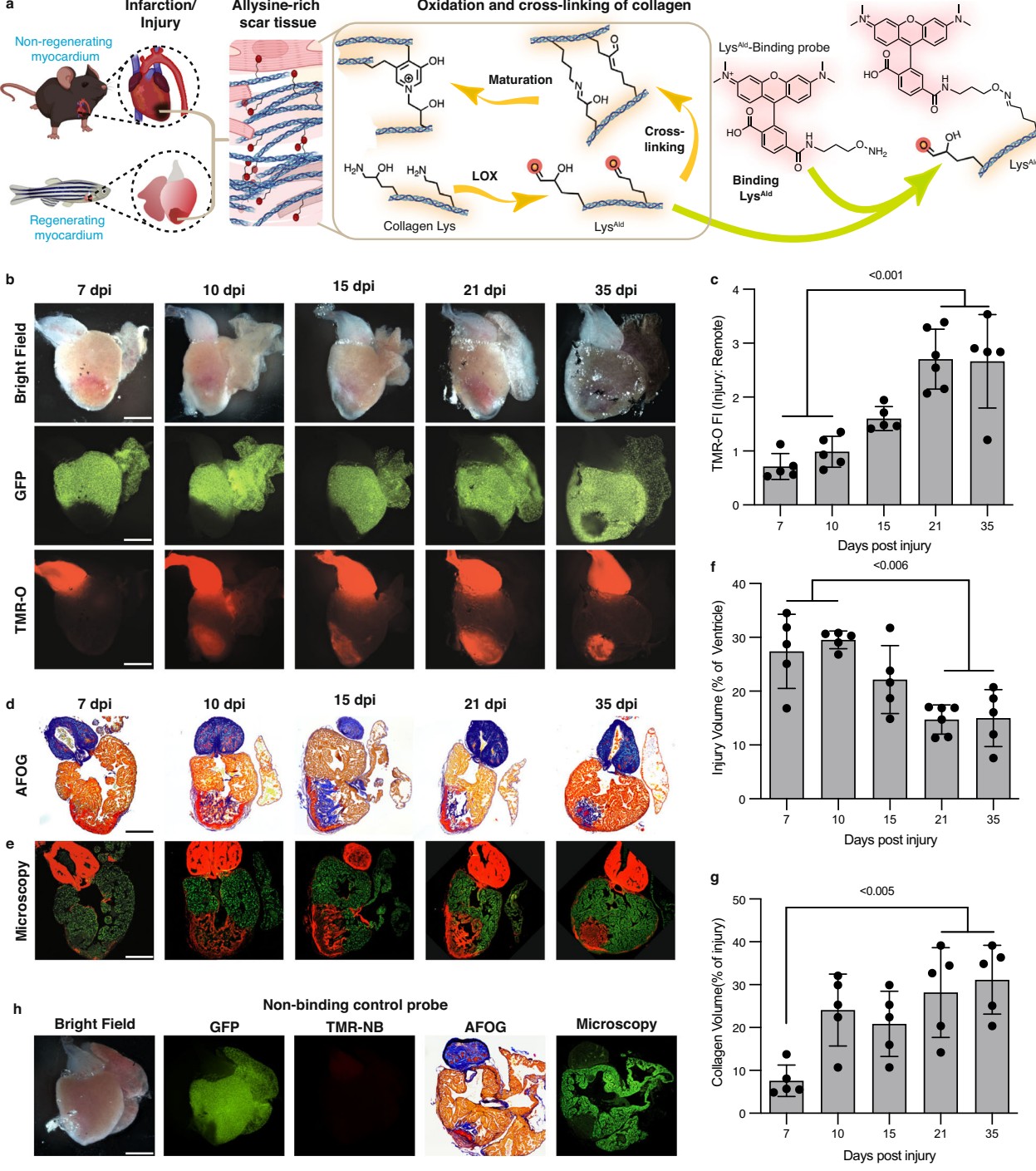

**Fig. 3 | Collagen oxidation and cross-linking in the injured zebrafish heart.**
Created in part with Biorender.com. **a** In the early stages of healing, a similar
process occurs in injured zebrafish and murine hearts: Lysyl-oxidase- (LOX)-medi-
ated oxidation of collagen in the wound converts lysine/hydroxylysine residues to
highly reactive aldehydes (allysine/hydroxyallysine, Lys$^{Ald}$), which form cross-links.
TMR-O binds to Lys$^{Ald}$, providing a readout of the initial steps in collagen cross-
linking. **b** Natural history of collagen oxidation and cross-linking in a zebrafish
cryoinjury model ($n = 5$ except for 21 dpi, $n = 6$). Adult zebrafish, with
cardiomyocyte-specific expression of GFP, were injected with TMR-O (2 nmol/g, IP)
and the hearts were excised and imaged 4 h after injection. The absence of GFP
delineates the area of injury, which shows a steady increase in TMR-O signal from
10–21 days post-injury (dpi). **c** TMR-O fluorescence intensity peaked between
21–35 dpi ($n = 5$ except for 21 dpi, $n = 6$). **d** AFOG-stained histology sections show

the collagen/elastin rich bulbus and collagen-rich healing injury at 10–35 dpi ($n = 5$
except for 21 dpi, $n = 6$). **e** Fluorescence microscopy of adjacent sections shows that
the TMR-O signal is localized to the collagen-rich areas of the injured heart ($n = 5$
except for 21 dpi, $n = 6$). **f** The total area of injured myocardium, quantified by
histology, decreased steadily between 10–35 dpi, consistent with the regenerative
capacity of the zebrafish heart ($n = 5$ except for 21 dpi, $n = 6$). (**g**) Collagen content
within the area of injury peaked 21–35 dpi ($n = 5$ except for 21 dpi, $n = 6$). **h** The
hearts of injured zebrafish injected with the non-binding probe TMR-NB (2 nmol/g,
IP) at 15 dpi showed no fluorescence signal in the heart or bulbus ($n = 3$). Data are
means ± SD of five-six measurements, where each data point represents one zeb-
rafish. *P*-values are shown where significant, one-way ANOVA with post hoc com-
parisons, two tailed. Scale bar = 300 μm. Source data are provided in the Source
Data file.

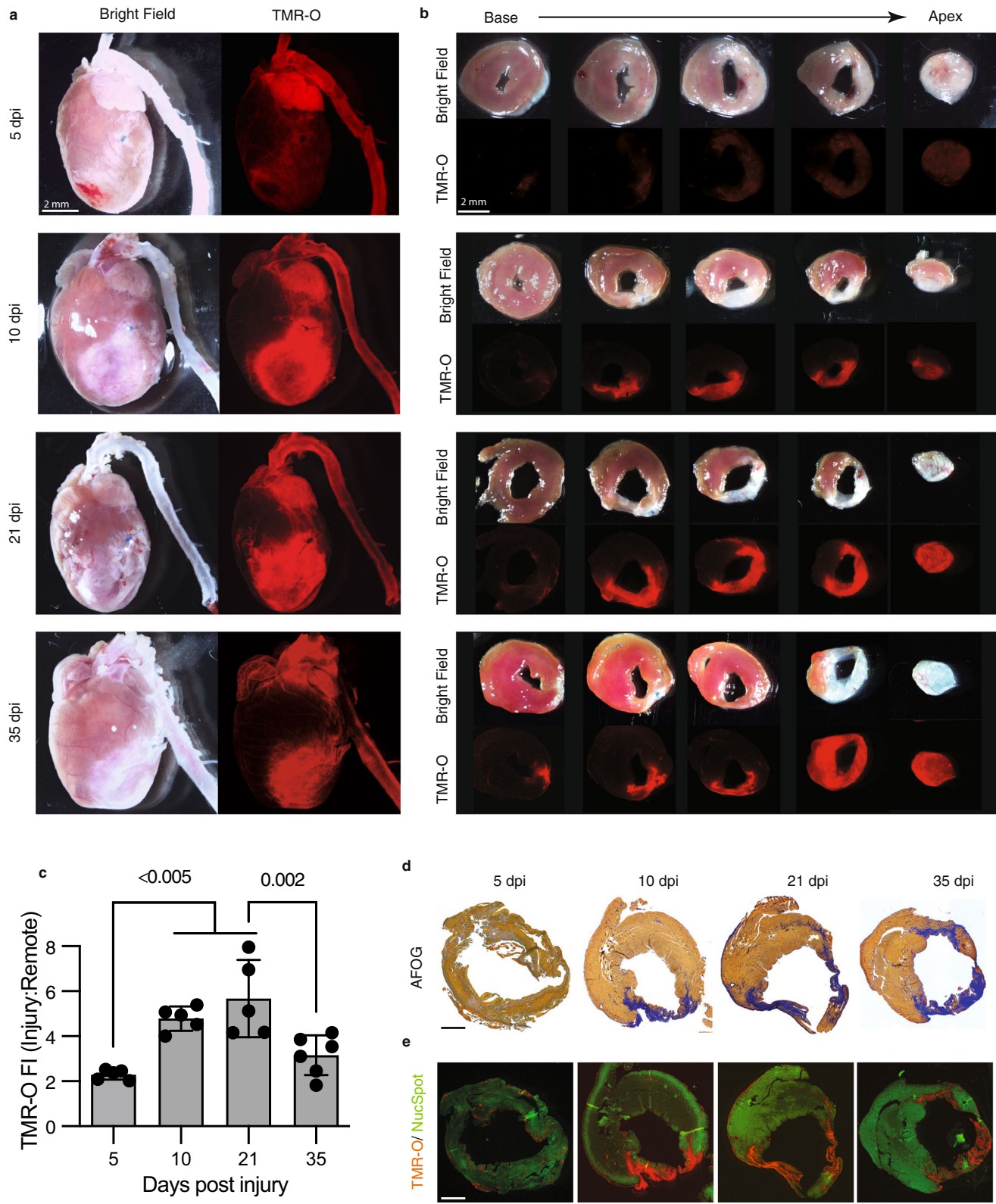

**Fig. 4 | Collagen oxidation and cross-linking in the infarcted mouse heart. a** The hearts of adult mice injected with TMR-O (2 nmol/g, IV) at 5–35 days after myocardial infarction (MI) are shown in their long axis. TMR-O fluorescence in the infarct is, qualitatively, already high at day 10, ($n = 5$ except for 35 dpi where $n = 6$). **b** Sectioning of the heart into 1 mm thick short axis slices confirms that the TMR-O signal is arising from the infarcted myocardium and is patchy but substantial by day 10 ($n = 5$ except for 35 dpi where $n = 6$). **c** Time course of TMR-O fluorescence in murine infarcts($n = 5$ except for 35 dpi where $n = 6$). The oxidation and cross linking of collagen is already significant by day 10, peaks at day 21 and subsequently decreases. Data are means ± SD of 5-6 measurements where each data point represents one mouse. Source data are provided in the Source Data file. *P*-values are shown where significant, one-way ANOVA with post hoc comparisons, two tailed. **d, e** AFOG-stained histology sections at the mid-ventricular level show that the collagen-rich injury area (blue) colocalizes well with the TMR-O signal on fluorescence microscopy ($n = 5$ except for 35 dpi where $n = 6$). Healthy myocardium is stained with a green, fluorescent nuclear stain (NucSpot). Scale bars = 1 mm.

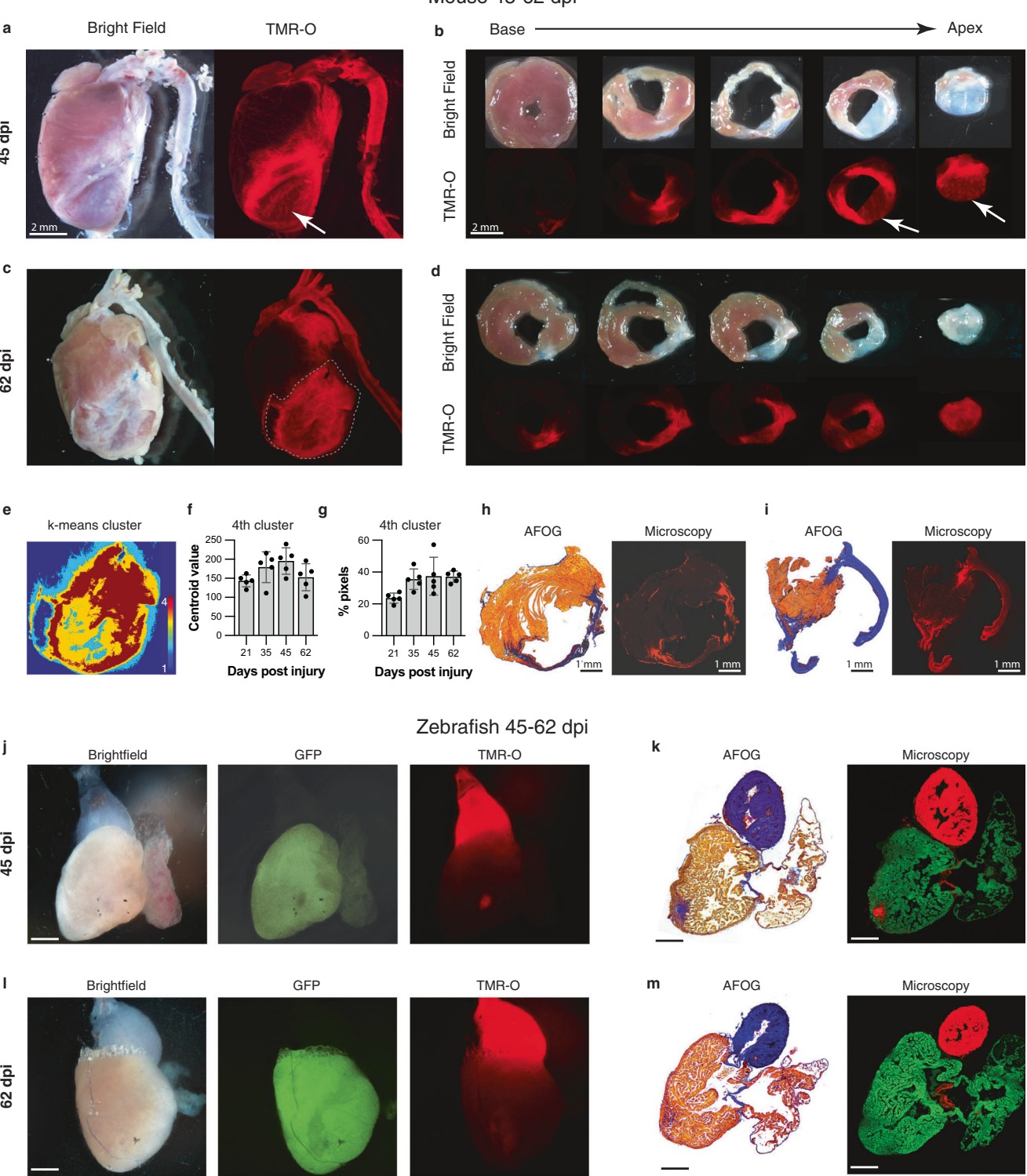

**Fig. 5 | Dynamic deposition and oxidation of new collagen in chronic infarcts.**
**a**, **b** Whole-heart and short axis section of a mouse heart 45 dpi. Some areas of the remodeled scar are structurally static with little TMR-O uptake (arrows, *n* = 5). However, other areas of the scar show substantial amounts of TMR-O uptake, consistent with the oxidation of newly deposited collagen. **c**, **d** Similarly, at day 62 some areas of the scar show high amounts of probe uptake, consistent with dynamic collagen deposition (*n* = 5). **e**–**g** K-means clustering (4 clusters) of the infarct zone in the whole-heart TMR-O images. **e** TMR-O clusters in a day 62 infarct zone, demarcated by the dashed lines in panel C. The cluster of highest TMR-O signal (4th cluster, brown) occupies a substantial portion of the infarct. **f**, **g** The centroid value and area of the 4th cluster did not change significantly over time (*n* = 5, values are plotted as the

average ± the standard deviation) one-way ANOVA, with Tukey's multiple comparisons, consistent with active/new collagen deposition in the infarcts at all stages of healing. Source data are provided in the Source Data file. **h**, **i** AFOG staining, and fluorescence microscopy confirmed the heterogeneity of TMR-O uptake in the day 45 and 62 murine infarcts (*n* = 5). **j**, **k** In injured zebrafish hearts (*n* = 5, scale bar = 300 µm.) most of the myocardium had been regenerated by 45 dpi and the size of the residual injury was small. However, within these small residual areas of injury substantial levels of TMR-O uptake were still seen. **l**, **m** Within 2 months of injury no evidence of myocardial injury or TMR-O uptake was seen in the zebrafish hearts (*n* = 5, scale bar = 300 µm.), consistent with complete resorption of the scars and successful regeneration of the myocardium. Scale bars in (**g**–**j**) are 300 µm.

periphery, remain highly dynamic with substantial deposition and cross-linking of new collagen.

By day 45 In the zebrafish hearts only very small areas of injury could be detected (Fig. 5j, k). However, these small injury zones remained dynamic and showed substantial uptake of TMR-O, consistent with ongoing collagen deposition and cross-linking, despite the greatly diminished area of injury (Fig. 5j, k). At 62 dpi, the zebrafish ventricle was completely regenerated, and little evidence of infarction could be detected by TMR-O fluorescence or by histology (Fig. 5l, m). These data demonstrate that the evolution of the fibrotic scars in zebrafish at days 45 and 62 after injury showed a very different pattern from that observed in the fibrotic scars in mouse hearts.

### Degradation-resistant cross-links develop in murine infarcts but not zebrafish infarcts

The formation of lysine-aldehydes is the first step in the cross-linking of collagen. This process is initiated by lysyl oxidase-mediated oxidation of lysine residues of the collagen telopeptide, the non-helical ends of the collagen triple helix (Fig. 6a). The lysine-aldehydes react with each other and other adjacent moieties to form the initial CCLs and stabilize the scar (Fig. 6b). However, these immature CCLs are unstable and can be hydrolized[28]. We aimed to determine whether the initial cross-links in injured zebrafish and murine hearts evolve into more mature and degradation-resistant forms. The maturation of collagen cross-links occurs by non-enzymatic processes and is considered to be spontaneous (Fig. 6b, c), but can be influenced by the exact peptide-sequence and glycosylation state of collagen[28]. The two most well-studied end-products of mature cross-linked collagen are pyridinoline (Pyd) and deoxypyridinoline (DPD) (Fig. 6d).

The quantification of Pyd and DPD levels in injured zebrafish and mouse hearts was performed using a previously validated HPLC method[29]. As both aromatic cross-links are fluorescent, this method does not require a labeling step for detection of the cross-links, and can therefore be carried out on small amounts of tissue. We assessed Pyd and DPD in whole zebrafish ventricles and the apical portion of the mouse infarcts. The amount of hydroxyproline (Hyp), a major amino acid in collagen and a well-established surrogate marker of collagen levels, was quantified in the initial tissue digests, which were then reconstituted to contain similar amounts of hydroxyproline/collagen (Supplementary Fig. S5). At 21 dpi, we found a prominent peak for both DPD and Pyd in infarcted mouse hearts but not in injured zebrafish hearts despite similar hydroxyproline concentrations in the analyzed samples (Fig. 6e). Pyd levels at 21 dpi in injured zebrafish and murine hearts, normalized to total hydroxyproline content are shown in Fig. 6f. No Pyd or DPD was detected in zebrafish tissue at any point after injury. In contrast, in murine infarcts, Pyd was always detectable and increased linearly with infarct age (Fig. 6g). DPD levels in the murine infarcts were substantially lower than Pyd levels (Fig. 6h, note scale difference vs. Pyd in panel G) and showed a Gaussian-like distribution as a function of infarct age. Collectively, these data show that fundamental differences in the development of mature, degradation-resistant, collagen cross links exist in injured zebrafish and murine hearts.

### Lysine hydroxylation in infarct collagen differs in murine and zebrafish hearts

The formation of both Pyd and DPD requires the conversion of at least one collagen lysine residue to hydroxylysine[9,30]. We hypothesized, therefore, that the lack of Pyd and DPD CCLs in the injured zebrafish hearts may be due to reduced hydroxylation of collagen lysine. The hydroxylation of collagen lysine residues occurs through the action of lysyl hydroxylases[10], and both hydroxylated and non-hydroxylated lysine residues in the telopeptide region of collagen undergo oxidation by lysyl oxidases to form (hydroxy)allysine [(OH)Lys$^{Ald}$] (Fig. 6a). TMR-O and other aldehyde-reactive probes bind to both allysine (Lys$^{Ald}$) and

hydroxyallysine (OH-Lys$^{Ald}$) and cannot provide a readout of lysine hydroxylation.

To quantify hydroxylysine levels in injured zebrafish and mouse hearts we used a well-established HPLC method[31,32], which involves a fast and quantitative derivatization reaction of amine groups with 9-Fluorenylmethyl Chloroformate (FMOC-Cl). Labeling of tissue hydrosylates with FMOC-Cl results in one fluorescent FMOC tag per amine group (Supplementary Fig. S6), which includes the alpha amine groups in all amino acids as well as any additional amines on amino acid sidechains. Following the labeling, an HPLC method with a gradient of three solvent systems is used to resolve all amino acids from one another in one chromatogram[31,32]. To account for differences in total tissue content between samples, aspartic acid (Asp), which showed a consistent concentration across all samples (Supplementary Fig. S7), was used as an internal standard. The chromatograms shown in Fig. 7 are normalized to the Asp peak such that the Asp peak height is at 100%. In zebrafish, the hydroxyproline (Hyp) peak in the chromatograms increased from 1–3 weeks after injury, but the hydroxylysine (OHK) peak remained unchanged (Fig. 7a and Fig. 7b). In the mouse infarcts, however, the increase in the Hyp peak from 1–3 weeks was accompanied by a significant increase in the OHK peak (Fig. 7c and Fig. 7d).

For quantitative analysis, the values of both Hyp and OHK were normalized to Asp levels in each sample. In the infarcted mouse hearts, significant increases in the amount of both Hyp and OHK were seen from 1–3 weeks after injury (Fig. 7e). In zebrafish, a significant increase in Hyp but not in OHK was seen between 1–3 weeks (Fig. 7f). Between weeks 1 and 3, Hyp and OHK levels in mice increased by 530% and 300%, respectively (Fig. 7g). The percent increase in Hyp in the zebrafish hearts was also substantial (170%) but the percent increase in OHK was minimal (8%) (Fig. 7g). Collectively, these data suggest that the limited hydroxylation of lysine residues in newly deposited collagen in the injured zebrafish heart impedes the formation of irreversible collagen cross-links (Pyd and DPD) in zebrafish infarcts.

## Discussion

The regenerative capacity of the human heart is extremely limited[33], making it particularly vulnerable to injury and infarction. A similar pattern is seen in all adult mammalian hearts, where injury results in a permanent fibrotic scar and the development of heart failure. A compelling need, therefore, exists to fully understand how the adult zebrafish resorbs the fibrotic scar and regenerates its heart after injury. Most of the focus of prior research in the zebrafish heart has been on the properties of the cardiomyocytes, such as their ploidy, that affect regeneration[34]. The nature of the fibrotic scar in the zebrafish heart, and whether this might play a role in its ability to regenerate, has been less studied.

The formation of a fibrotic scar involves the deposition of extracellular collagen fibrils, which must be cross-linked to provide mechanical stability to the scar[30]. The extent of cross-linking has long been presumed to be central to collagen stability and its resistance to degradation[8,35–37]. We set out to test this hypothesis in zebrafish and murine models of cardiac injury by developing optical probes that bind to aldehydes on collagen such as allysine (Lys$^{Ald}$). The small size of the zebrafish heart (<1 mg) makes it challenging to characterize collagen cross-linking with available methods. The amount of infarcted tissue is insufficient to detect/characterize with MRI and many standard chemical techniques. Fluorescence imaging provides both extremely high sensitivity and spatial resolution and is well suited for imaging the zebrafish heart. In addition, probe kinetics such as binding and dissociation rates can be modulated by the overall structure and active moiety in the fluorescent probe. We therefore developed a library of aldehyde-binding 6-carboxy tetramethyl rhodamine probes of which TMR-O showed the best properties for imaging of the zebrafish heart.

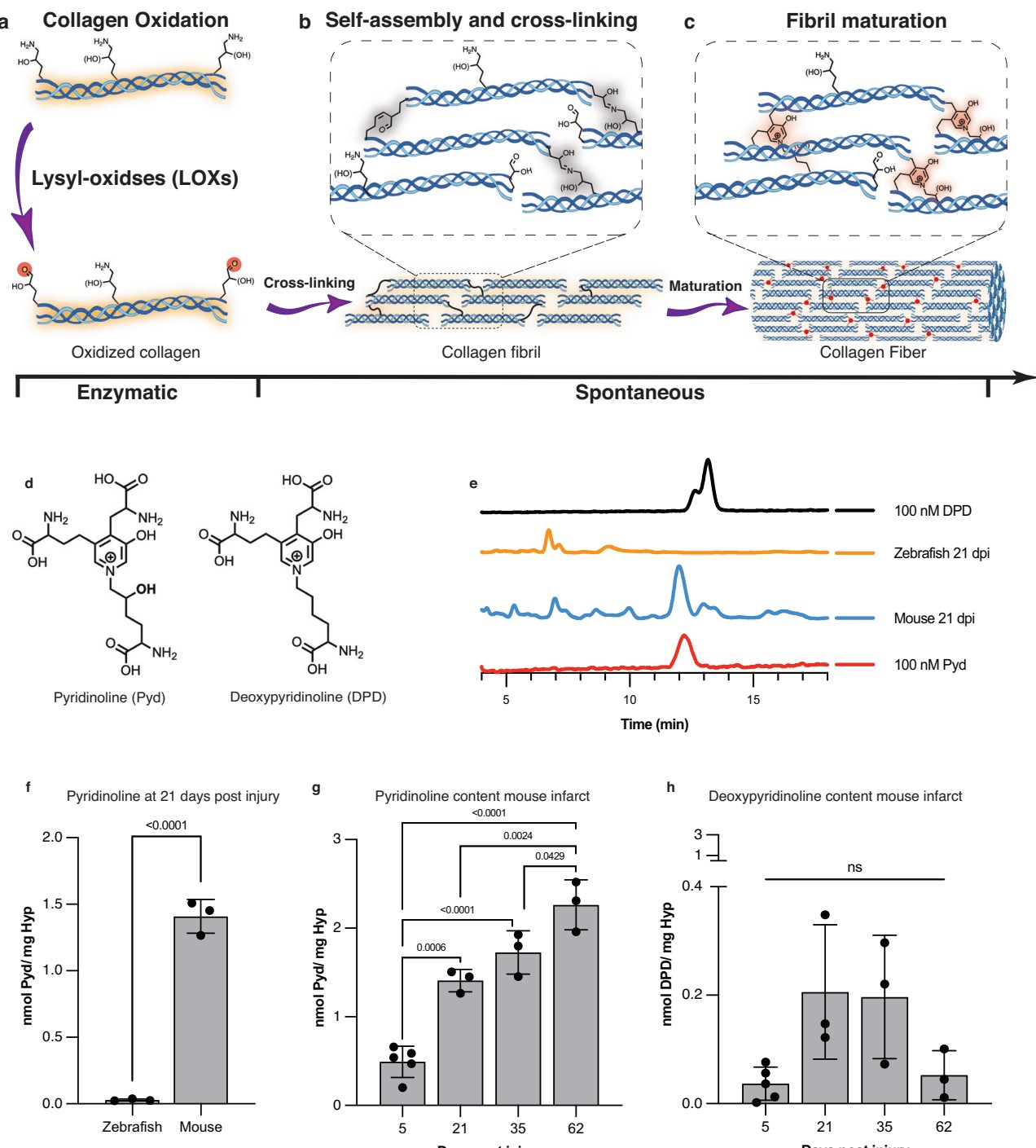

**Fig. 6 | Lysine aldehyde cross-links mature into degradation-resistant final products in infarcted mouse hearts but not in zebrafish hearts.** Created in part with Biorender.com. **a** Enzymatic oxidation of telopeptide [hydroxy]lysine residues of collagen by Lysyl oxidase (LOX) to form [hydroxy]allysine aldehydes is a prerequisite of collagen cross-linking and fibril formation. **b** Subsequent self-assembly of collagen molecules is spontaneous and is stabilized by intramolecular reactions of [hydroxy]allysine groups with one another or with helical [hydroxy]lysine residues. **c** Initial allysine cross-links further rearrange and react with additional [hydroxy]allysine or [hydroxy]lysine groups to form stable pyridinium-containing cross-links. **d** Chemical structures of pyridinoline (Pyd) and deoxypyridinoline (DPD), the mature degradation-resistant products of collagen cross-linking. **e** HPLC traces of a solution of DPD (100 nM, top, black) and Pyd (100 nM, bottom, red) juxtaposed with traces of solutions of hydrolyzed mouse or zebrafish hearts excised 21 days after injury. All zebrafish and mouse heart hydrolysates were

prepared to contain similar amounts of hydroxyproline (3–8 μg/mL).
**f** Quantification of Pyd in infarcted mouse and zebrafish hearts 21 days post injury (dpi) shows no detectable Pyd levels in zebrafish hearts despite substantial hydroxyproline content in the analyzed tissue ($n = 3$, t-test). Similarly, DPD could not be detected in the HPLC traces of the zebrafish hearts. **g** In infarcted mouse hearts, Pyd levels increase with infarct age consistent with scar maturation ($n = 3$ except for 5 dpi where $n = 5$, one-way ANOVA with Tukey's post hoc comparison). **h** Levels of DPD in infarcted mouse hearts peak earlier after infarction and are lower than Pyd ($n = 3$ except for 5 dpi where $n = 5$, one-way ANOVA with Tukey's post hoc comparison, note the scale difference between panels g and h). Data are means ± SD of 3–5 independent measurements where each data point represents one animal. T-test One-way ANOVA with Tukey's post hoc comparison Source data are provided in the Source Data file.

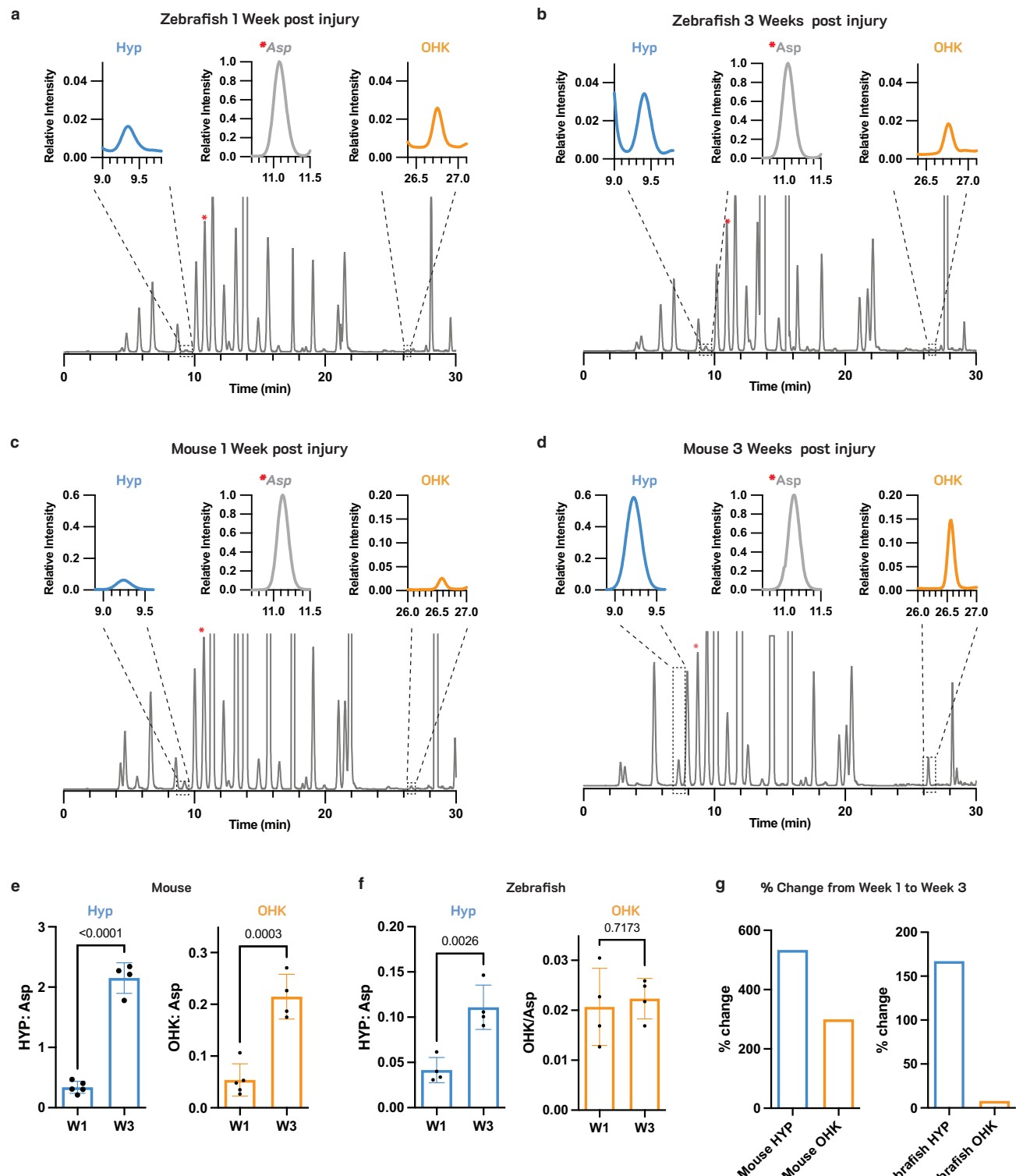

**Fig. 7 | Hydroxylation of lysine in newly deposited collagen differs markedly in infarcted murine and zebrafish hearts.** HPLC traces of the amino acids in infarcted zebrafish (**a**, **b**) and mouse (**c**, **d**) hearts at 1 and 3 weeks after injury. The expanded regions in the insets are for the hydroxyproline (Hyp) and hydroxylysine (OHK) peaks, at 9.3 min and 26.7 min respectively, and the Aspartic acid (Asp) peak at 11.3 min to which the chromatogram is normalized. Aspartic acid (Asp) was quantified and used to normalize Hyp and OHK values to account for total tissue content in each sample. The Asp peak is denoted with a red asterisks (*) in the full HPLC chromatograms. Quantification of Hyp:Asp and OHK:ASP in mouse (**e**, $n = 5$ for W1, $n = 4$ for W3) and zebrafish (**f**, $n = 4$) hearts at 1 and 3 weeks after injury. Hyp:Asp increases significantly in both zebrafish and mice, while OHK:Asp increases significantly in mice but not zebrafish. **g** Comparison of the percent change of Hyp and OHK from week 1 to week 3 after injury in mice and zebrafish hearts. Despite a marked increased in Hyp from weeks 1–3, indicative of new collagen deposition, little change in OHK is seen in the zebrafish hearts. Statistical comparison by unpaired $t$-test, $n = 4$–5 per sample values are plotted as the average ± the standard deviation. Source data are provided in the Source Data file.

The fast condensation rate of TMR-O with aldehydes and the slow hydrolysis of the resulting oxime results in a much higher accumulation of the probe on Lys^Ald-rich proteins and tissues compared to the other analogs in the library. For the other Lys^Ald-binding analogs, we found that the off-rate (hydrolysis) had the greatest effect on on-target accumulation. The clearest evidence of the dominating effect of hydrolysis is observed in comparing TMR-O and TMR-LHZ which have near identical structures and near identical on-rate reactivity. However, with TMR-O, the slow hydrolysis of the oxime bond results in a much higher accumulation on target tissue in vivo[38]. The slow hydrolysis of oximes compared to hydrazones is a known phenomenon resulting from the electron withdrawing nature of the oxygen in oximes compared to nitrogen in hydrazones[39,40]. This so called "alpha nucleophile" dictates the rate of hydrolysis, as we have recapitulated in our study. The rate constant of TMR-O was 27-fold higher than the only commercially available aldehyde-binding fluorochrome (TMR-HZD), was significantly more soluble, and also showed no evidence of impurities on HPLC. Using TMR-O, we also showed that collagen oxidation in the early phases of healing infarcts is similar in the mammalian and zebrafish heart but that fundamental differences evolve as the scar matures. Specifically, we show that mature and degradation-resistant collagen cross-links, such as Pyd and DPD, form in murine but not zebrafish hearts after injury. This difference could play a key role in the ability of the zebrafish heart to resorb fibrotic scars and replace them with regenerated myocardium. Interestingly, the level of DPD in the murine infarcts was substantially lower than Pyd and showed a Gaussian pattern over time, unlike the steady and linear increase in Pyd. This difference between Pyd and DPD production suggests that DPD cross-links may form in portions of the infarct that become static over time while Pyd cross-links in those portions of the infarct that remain more dynamic with ongoing collagen deposition.

The absence of Pyd and DPD in zebrafish hearts is likely a consequence of the limited hydroxylation of lysine in newly deposited collagen in the injured zebrafish heart. The hydroxylation of lysine precedes collagen oxidation and is required for the formation of Pyd and DPD[41]. In the absence of lysine hydroxylation, collagen in the injured zebrafish heart cannot form the degradation resistant Pyd and DPD cross-links and persists in a form that is more amenable to breakdown and removal. To the best of our knowledge, this is the first study to evaluate the degree of collagen oxidation, hydroxylation and cross-linking in zebrafish infarcts and detect biophysical differences in the properties of collagen between mammalian and zebrafish hearts.

The initial step in cross-linking involves the oxidation of lysine/hydroxylysine to form highly reactive aldehydes, which undergo condensation reactions with each other and with other adjacent functional groups. The presence of Lys^Ald in tissue has been therefore presumed to be an indicator of recently deposited collagen and a sign of active cross-linking[42,43]. Experience with allysine-binding probes in other tissues bears this out[11-13,44-47]. In the liver and lung the uptake of MRI-detectable Lys^Ald-binding probes is seen only during the acute phase of injury, and ceases when the fibrotic stimulus is removed[12,44]. The uptake of TMR-O in 2 month-old murine infarcts suggests, therefore, that collagen deposition in portions of these scars remains highly dynamic and active. Rather than infarct expansion being due only to mechanical stretching[26,48,49], our data suggest that mature scars in the mammalian heart may undergo expansion in large part due to the deposition of new collagen. This also implies that the therapeutic window to modulate the properties of infarcted myocardium may be longer than previously thought.

In conclusion, a high-precision molecular imaging probe was developed in this study to characterize the oxidation and cross-linking of collagen in injured zebrafish and mouse hearts. While collagen oxidation and cross-linking are initially fairly similar in injured zebrafish and murine hearts, significant differences in the maturation of the collagen-cross links evolve over time. The absence of degradation-resistant cross-links in the infarcted zebrafish heart may play an important role in allowing it to resorb fibrotic scars and replace these with healthy regenerated myocardium.

## Methods

### Synthesis and Characterization of compounds

Detailed synthetic procedures and characterization of compounds can be found on the accompanying Supporting information file.

**Materials and instruments.** NMR spectra were recorded on a JEOL ECZ 500 R 11.7 T NMR system equipped with a 5 mm broadband probe ($^1$H: 499.81 MHz, $^{13}$C: 125.68 MHz) using NMRDelta 6.0.0 software. $^1$H NMR spectra were analyzed using MestreNova 14.3.1spectral shifts are reported as singlet (s), doublet (d), triplet (t), quartet (q) or multiplet (m). UV-Vis spectra were recorded on a SpectraMax M2 spectrophotometer using cuvettes with a 1 cm path length. MilliQ purified water was used for all reactions and experiments where water is the solvent. 6-carboxytetramethyl rhodamine was purchased from Fisher and used without purification. TAMRA hydrazide, 6 isomer was purchased from Lumiprobe and used without further purifications. Tert-Butyl 3-aminopropoxycarbamate was purchased from AmBeed and used without further purification. All other chemical supplies and reagents were purchased from common commercial suppliers and used without purifications unless otherwise noted.

LC-MS analysis was carried out on an Agilent 1260 system (UV detection at 220, 254 and 550 nm) coupled to an Agilent Technologies 6130 Quadrupole MS system. HPLC with fluorescence detection was acquired on an Agilent 1260 with binary pump, autosampler, multiwavelength detector, thermostatted column compartment, and vacuum degasser and a fluorescence detector.

### In vitro characterization

**Determination of forward reaction rates.** Reaction rates of aldehyde-binding compounds with butyraldehyde were determined using high performance liquid chromatography (HPLC) with fluorescence detection (excitation 545 nm, emission 566 nm). First, an HPLC trace at $t = 0$ min for 10 μM solution of TAMRA probes (0.4 mL pH 7.40 PBS) was acquired. Then, 9 μL of a 5 mM solution of butyraldehyde in PBS was added for a final [butyraldehyde] of 110 μM, and spectra were collected every 7–10 min. Integrations for starting material and product peaks were fit to the following integrated rate equation for a second order reaction as previously described:

$$\ln \frac{[\text{Probe}]\,[\text{Butyraldehyde}]_{t0}}{[\text{Butyraldehyde}]\,[\text{Probe}]_{t0}} = k\left([\text{Probe}]_{t0} - [\text{Butyraldehyde}_{t0}]\right)t$$

$$(1)$$

where [Probe] is the concentration of the aldehyde-reactive probe at any timepoint (t), [Butyraldehyde] is the concentration of butyraldehyde at any given timepoint (t) [Butyraldehyde]$_{t0}$ is the initial butyraldehyde concentration, and [Probe]$_{t0}$ is the initial concentration of probe. This equation can be rearranged to give:

$$\ln \frac{[\text{Probe}]\,[\text{Butyraldehyde}]_{t0}}{[\text{Butyraldehyde}]\,[\text{Probe}]_{t0}} \times \frac{1}{\left([\text{Probe}]_{t0} - [\text{Butyraldehyde}_{t0}]\right)} = kt$$

$$(2)$$

Because [Butyraldehyde] is in 10-fold excess to probe, we can assume it does not change significantly throughout the reaction, and [Butyraldehyde] μμ ≈ [Butyraldehyde]$_{t0}$ and the equation can be

simplified to:

$$\ln \frac{[\text{Probe}]}{[\text{Probe}]_{t0}} \times \frac{1}{\left([\text{Probe}]_{t0} - [\text{Butyraldehyde}_{t0}]\right)} = kt \qquad (3)$$

Plotting

$$y = \ln \frac{[\text{Probe}]}{[\text{Probe}]_{t0}} \times \frac{1}{\left([\text{Probe}]_{t0} - [\text{Butyraldehyde}_{t0}]\right)} \times x$$

Where x as the time in minutes gives a straight line with the slope as the rate constant in $M^{-1}min^{-1}$

For each probe, reaction rate was measured at least three times, and the rate constant is reported as the mean ± standard deviation between values.

**Determination of hydrolysis or reversibility kinetics.** Hydrolysis of the product between aldehyde-reactive probes and butyraldehyde was monitored using high performance liquid chromatography (HPLC) with fluorescence detection (excitation 545 nm, emission 566 nm). First a 10 µM solution of TAMRA probes (1 mL pH 7.40 PBS) was incubated with 100-fold equivalents of butyraldehyde (1 mM) at 37 °C overnight. The resulting solution was aliquoted into 4 vials of 250 µL each and freeze-dried to remove the excess butyraldehyde and isolate the product. To the resulting pink powder was added 250 µL of water, and the sample was immediately subjected to HPLC. Samples were run every 7 min to observe reaction reversibility.

**Binding to BSA-Ald.** BSA-Ald was prepared as previously described: A solution of bovine serum albumin (BSA, 500 mg) was dissolved in PBS (10 mL, pH 7.4) along with sodium ascorbate (100 mg). A freshly prepared solution of ferric chloride (50 µL, 10 mM) was then added to the stirring BSA solution, and the mixture was stirred overnight at room temperature. The mixture was concentrated using a 5 kD molecular weight cut off filter (MWCO), and the protein was purified using PD-10 Sephadex G25 desalting columns with PBS as eluent. Protein concentration was determined with BCA Protein Assay Kit, (Thermo Scientific) and aldehyde concertation was determined using Protein Carbonyl Content Assay Kit (Sigma-Aldrich). Aldehyde concentration was determined to be 4 mol aldehyde per mol protein. BSA-Ald was diluted in PBS (pH 7.4) to a concentration of 25 µM protein, 100 µM aldehyde (1.6 mg BSA-Ald/mL).

For binding studies, 7 µL of a 1 mM solution of TAMRA-based probes was added to 0.7 mL of BSA-Ald stock for final concentration of 10 µM TAMRA and 100 µM aldehyde. Solutions were then incubated at 37 °C to 3 h. UV-Vis absorption spectra were recorded for each solution (400 nm–700 nm), and the solutions were then passed through a 10 kD MWCO filter by centrifugation. The filter was washed once with an additional 0.7 mL of PBS. UV-Vis spectra of both fractions were collected, and concentration of unbound probe was calculated based on reported molar extinction coefficient of 6-TAMRA at 546 nm of 89,000 $L\,mol^{-1}cm^{-1}$.

For blocking experiments, 2 mL of a solution of methoxylamine hydrochloride in PBS (10 mM, pH 7.4) was added to a solution of BSA-Ald (10 mL, 50 µM protein) and an additional 8 mL of PBS was added. This solution was incubated for 3 h at 37 °C. Binding experiments were then carried out as described above. For methoxylamine solution, care was taken to ensure neutral pH of the solution before adding to BSA-Ald. In some cases, adjusting the pH to 7.40 with 6 M NaOH was necessary.

## Animal and tissue studies

**Materials and equipment.** Ex-vivo fluorescence images were obtained on Nikon SMZ-18 stereomicroscope with coolLED GFP ET and DS Red HC filters. Images were captured using DS-Ri2 camera with NIS-Elements BR 5.30.02 64-bit software. Manual z-stack images were merged using Zerene Stacker version 1.04. ImageJ[50] version 1.53 K was used for analysis of ex-vivo images of whole hearts and axial slices of mouse hearts.

Nucspot green fluorescence nuclear stain was purchased from VWR and used per manufacturer instructions. Multi-color fluorescence imaging of histological sections was done using a Zeiss LSM900 confocal microscope with a 40 x/1.2 water immersion objective. QuPath software version 0.5.1 was used for analysis histological sections. Pyridinoline and deoxypyridinoline were purchased from Toronto Research Chemicals and reconstituted in water to (1 mM). TSKgel™ ODS-80Tm HPLC Column was purchased from Supelco.

**Animal models.** Zebrafish were produced, grown, and maintained according to standard protocols approved by the Institutional Animal Care and Use Committees of Massachusetts General Hospital. Ethical approval was obtained from the Institutional Animal Care and Use Committees of Massachusetts General Hospital. For experiments with adult zebrafish, animals ranging in age from 4–10 months were used. Approximately equal sex ratios were used for experiments. Adult density was maintained at 4–5 fish/L for all experiments in Aquarius racks and fed twice per day. Water temperature was maintained at 28 °C. The published strains used in this study are Tg(cmlc2:H2B-EGFP)[fb501Tg] where specific expression of GFP is in the histones of cardiomyocytes[24], and Tg(cmlc2:nls-GFP)[fb18Tg] with cardiomyocyte-specific nuclear expression of GFP[25]. Cryoinjury of the ventricle to mimic myocardial infarction was carried out as previously described[6].

Mouse model of Myocardial infarction: C57Bl/6 J mice of approximately equal sex ratios, 10 weeks old and weighing at least 20 g were obtained from Charles River laboratories (Wilmington, MA, USA). All experimental procedures were approved by the Institutional Animal Care and Use Committee (IACUC). All mice were maintained on 12-h alternating light and dark cycles, with ambient temperature at 25 °C and humidity at 30% and all mice had free access to water and food. MI was achieved by the permanent ligation of the left coronary artery as previously described[51]. MI procedures were conducted by the Cardiovascular Physiology Core at the Mass General Brigham.

**Selection of time points for ex-vivo imaging.** Probe injection was performed intravenously (IV) into the tail vein in mice, and intraperitonially (IP) in zebrafish. IP injections rather than retro-orbital IV injections were used in zebrafish to limit inconsistencies in probe delivery. However, IP injections are known to result in slower entry of compounds into the circulation and, therefore, a longer half-life for elimination from the blood[52]. We aimed to image TMR-O, and the other probes in the study, once all unbound probe (non-specific background signal) had been eliminated from the blood. To characterize the elimination of the probes after IV (mouse) and IP (zebrafish) injection, non-conjugated (non-targeted) 5-carboxy tetramethylrhodmine (TAMRA) was injected (2 nmol/g) into naïve animals. The animals were euthanized at 0.5, 1, 1.5, 3 and 5 h after injection and hearts and aorta were imaged ex vivo. TAMRA was completely cleared from the myocardium and bulbus/aorta within 4 h post IP injection in zebrafish and 1.5 h post IV injection in mice. These time points (4 h in zebrafish, 1.5 h in mice) were, therefore, used in all study experiments.

**Ex-vivo fluorescence imaging.** Zebrafish at different time points post cryoinjury were anesthetized in tricaine, weighed, and injected interperitoneally with 2 nmol/g of the TAMRA-based fluorescent probes. The fish were then allowed to recover for 4 h. Animals were then

euthanized, and the heart and bulbus were excised, washed in heparinized PBS, and placed in a solution of 0.1 mM KCl for imaging.

Mice were anesthetized with isoflurane (2%) and injected with TMR-O (2 nmol/g) via a tail-vein catheter. Mice were euthanized 90 min after injection, and the hearts and aortas were excised and washed in heparinized PBS. Following imaging, the infarcted region of the heart was then cut into 5–6 × 1 mm-thick slices and imaged again before fixation.

**Image analysis.** To account for potential concentration variations in the formulated dose injected into each zebrafish, a calibration phantom using the same formulation of the probe was imaged with the bulbus in each case. The signal in each bulbus was normalized by the signal in its corresponding calibration phantom.

For zebrafish hearts, regions of interest (ROIs) were defined based on the brightfield and GFP- fluorescence images for the ventricle, bulbus, and injured ventricle. For mouse hearts ROIs were defined based on the brightfield images for the normal ventricle, aorta, and injured ventricle. The mean fluorescence intensity of the injury area was normalized to the intensity in the aorta/bulbus or the normal myocardium.

Histological analyses: Samples were fixed overnight at 4 °C in 4% PFA, embedded in paraffin and sectioned following conventional histological procedures. Acid fuchsin-orange G (AFOG) stain was used to detect fibrotic tissue. Muscle, fibrin/cell debris and collagen were stained brown-orange, red and blue, respectively. Slides were imaged without staining to assess distribution of TAMRA dyes using a Zeiss LSM900 point scanner confocal equipped with four laser lines (405,488,561 and 640 nm) and two detectors and 40x/1.1 Water immersion objective.

For k-means cluster analysis, a region of interest (ROI) was defined to include the entire visible area of injury. Segmentation of the TMR-O signal in this ROI was performed using k-means clustering (ImageJ) based on the brightness of the TMR-O signal. The fourth (brightest) cluster was presumed to contain pixels with the highest concentration of freshly oxidized (and therefore freshly deposited) collagen. The percentage of the overall ROI occupied by pixels in the fourth cluster and the average intensity of this cluster were then determined.

**Quantification of injury volume from histological sections.** Histological slides of zebrafish hearts covering the entire volume of the heart, and confocal fluorescence images of unstained, unbleached sections show significant GFP fluorescence of GFP-positive cardiomyocytes, and TMR-O fluorescence in the bulbus and regions on the injury. Separate ROIs were drawn for the entire ventricle in each tissue section, and for the injury region within the same tissue sections. The volume of injury was then calculated by summing the values for the area of all the injury ROI's and dividing by the sum of the values of the area of entire ventricle ROI's.

**Cross-link and hydroxyproline analysis.** The apical most slice of mouse hearts and the entire ventricle of fish hearts were digested in 100–200 μL of 6 M HCl at 110 °C overnight. A portion of the digest (5–10 μL) was neutralized with an equal volume of 6 M NaOH and analyzed for hydroxyproline as previously reported. A second portion of the digest (50–100 μL) was evaporated to dryness by freeze-drying. The dried digest was then reconstituted in water to achieve hydroxyproline concentration of 40–80 μg/mL (based on hydroxyproline analysis). This solution was then subjected to HPLC analysis for DPD and Pyd as previously reported[29].

**Analysis of hydroxylysine, aspartic acid and hydroxyproline by labeling with 9-Fluorenylmethyl Chloroformate.** The same tissue hydrosylates used for the cross-link and hydroxyproline analysis were used in this analysis too. A portion of the digest (5–10 μL) was neutralized with an equal volume of 6 M NaOH, then diluted with borate acid buffer and derivatized with 9-Fluorenylmethyl Chloroformate (FMOC-Cl) as previously described[31,32]. In cases where the identity of the peak was unclear, the samples were run with and without a spiked-in standard of the labeled amino acid in question and HPLC traces were compared before and after the addition of the standard.

**Statistical analyses.** Statistical analyses were completed using GraphPad prism version 10.1.1 using one-way ANOVA with post hoc Tukey's multiple comparisons test. P-values are typically not listed for each comparison, rather statistical significance for a group is indicated with the highest p-value. For example, for Fig. 3E, comparison between 7 and 21 dpi and 7 and 35 dpi as well as 10 and 21 dpi and 10 and 35 dpi were all statistically significantly different, and figure indicates that both 7 and 10 dpi are statistically significantly different from 21 and 35 dpi. In cases where only two groups are compared (i.e. Fig. 7c, d), an unpaired t-test was performed to evaluate significance.

## Reporting summary
Further information on research design is available in the Nature Portfolio Reporting Summary linked to this article.

## Data availability
The data associated with this study are provided in the main manuscript and in the Source Data file. Complete details on the synthesis and characterization of all experimental compounds are provided in the Supplementary Materials file. Data are also available from the authors upon request. Source data are provided with this paper.

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

## Acknowledgements

This work is funded in part by the following grants from the National Institutes of Health: K01HL155237 (EAB), R01HL164749 (JMGR), R01DK121789 (PC), R33HL154125 (P.C.), S10OD032138 (P.C.), and S10OD025234 (P.C.). EAB is also funded by the Mass General Brigham ECOR CDI Physician Scientist Development Award. D.B. received support from the Wallonie Bruselles International Postdoctoral Fellowship. J.M.G.R. is also supported the American Heart Association (19CDA34660207), the Corrigan Minehan Foundation (SPARK Award), and the Hassenfeld Foundation (Hassenfeld Research Scholar Award).

## Author contributions

E.A.B.—study conception, study design, probe development, probe synthesis, data acquisition, data analysis, data interpretation, manuscript preparation, study funding. D.B.—study design, preparation of animal

models, data interpretation, editing of manuscript. S.J.R.—data acquisition, histology, microscopy, editing of manuscript. S.T.—histology, microscopy, editing of manuscript. B.A.—histology, microscopy, editing of manuscript. B.M.—preparation of animal models, data acquisition, editing of manuscript. H.M.—probe synthesis, editing of manuscript. Y.Z.—preparation of animal models, editing of manuscript. A.A.—preparation of animal models, editing of manuscript. P.C.—study design, probe development, data interpretation, editing of manuscript. J.M.G.R.—study conception, study design, preparation of animal models, data interpretation, editing of manuscript. D.E.S.—study conception, study design, data analysis, data interpretation, manuscript preparation, study funding, study supervision

## Competing interests

E.A., H.M., and P.C. are inventors of a filed patent based on the work here (Molecular probes for in vivo detection of aldehydes. PCT/US2022/072310). P.C. has equity in and is a consultant to Collagen Medical LLC, has equity in Reveal Pharmaceuticals Inc., and has research support from Pliant Therapeutics, Takeda, and Janssen. The remaining authors declare no competing interests.
