## [Peer Review File · Nature Communications]

Dynamics of Collagen Oxidation and Cross Linking in Regenerating and Irreversibly Infarcted MyocardiumREVIEWER COMMENTS

Reviewer #1 (Remarks to the Author):

Akam and colleagues developed a library of fluorescent probes to quantify collagen oxidation and cross-link formation, and then used them to study the dynamics of collagen cross-linking in mouse and zebrafish hearts after injury. They found that while the initial oxidation and cross-linking of collagen were similar in both species, mature degradation-resistant cross-links developed in infarcted mouse hearts but not in injured zebrafish hearts. This suggests that the absence of mature cross-links in zebrafish hearts may facilitate their ability to regenerate. Overall, the experimental design is well done and innovative, providing a new method to measure collagen oxidation and cross-link formation in hearts *in vivo*. It would be much improved if the authors provide any mechanistic insights on mature cross-links between zebrafish and mice. With that said, here are my suggestions and concerns:

1. In Figure 1, the structure and the on-rates for reaction of TMR-O and TMR-LHZ were similar, why the extent of product formation is different?
2. In Figure 3, the signal of TMR-O peaked at 21 days post injury, which means collagen oxidation and crosslinking peaked at this time point. Why the collagen volume didn't change from 21 to 35 days post injury?
3. In Figure 5, the study needs more numbers of mouse and zebrafish hearts, and the current data may limit the generalizability of the findings.
4. The study primarily present a method on detecting the dynamics of collagen oxidation and cross-links, but have no any mechanistic insights into the underlying processes, such as related gene expression and function on the context of collagen maturation.
5. The products of pyridinoline and deoxypyridinoline can be detected even without the aid of TRM-O. Ares there any specific small molecules that can be used to remove pyridinoline and deoxypyridinoline for treating CCL?
6. In line 136, TMR-HDZ should be TMR-HZD?

Reviewer #2 (Remarks to the Author):

This manuscript by Akam et al., develops new fluorescent labelling tools to assess the formation of collagen cross links following models of cardiac injury in mice and zebrafish. The study starts by assessing labelling kinetics of a library of probes *in vitro* and then concentrates on one probe, TMR-O, in models of MI in non-regenerative adult mice and regenerative adult zebrafish. The study is original and timely and the comparison between two different models adds considerable strength. The findings will be of interest to a broad range of investigators working in regenerative medicine and matrix biology. The studies appear well done and the results are highly interesting, particularly that early cross linking appears very similar between the two models and the fact that adult mouse collagen crosslinking appears dynamic over a longer period of time that previously thought.

I have a number of queries:

1 – On line 110 the authors state that the butyraldehyde, used as a target aldehyde for the *in vitro* investigations in Figure 1, has a “similar structure to the *in vivo* aldehyde target allysine.” How similar are they? How applicable is this data to the *in vivo* findings? I appreciate that the TMR-O probe performs best in all investigations but can more be learnt from investigating the other probes *in vivo* considering their different observed on and off rates?

2 – Can the authors comment on why different delivery methods were chosen for the mice (intravenous) and the zebrafish (intraperitoneal). Could these different methods result in different accumulation kinetics?

3 – Following on from this the authors describe different labelling timelines for mice and fish (90 mins vs 4 hours, respectively – data in Figures 3 and 4) but on line 314 say that the labelling in mice was “substantially higher than in the zebrafish.” Can these data really be compared? I think a direct quantitative comparison would be needed to make this conclusion.

4 – On line 396 the authors state “The maturation of collagen cross-links occurs by non-enzymatic processes and is considered to be spontaneous (Fig. 6b -c), but can be influenced by the exact peptide-sequence and glycosylation state of collagen.²⁸” Have the authors investigated differences in the peptide sequence and glycosylation state of collagens between mice and zf as a way to explain their findings?

5 – On line 409 and in the legend of Figure 6 the authors say that there are “similar” levels of hydroxyproline in the zf and mouse samples, is there a way to show this data? Or for these levels to be more precise? This would aid in the robustness of this data.

Minor Points for clarification:

Figure 5 legend, graphs in f and g – is each point one mouse? The legend says “did not change significantly over time”. How were the statistical tests done?

Line 391 – “resides” should be “residues”?

Line 392 – insert “other” after “each”

Reviewer #3 (Remarks to the Author):

The manuscript of Akam et al. presents the synthesis and characterization of TMR-O, a fluorescent compound for the staining of allysines (a marker of actively cross-linking collagen). The compound measures to have superior reactivity and stability compared to the commercially available TMR-HZD. With the TMR-O compound the time course and distribution of allysine are quantified in MI models in mouse and zebrafish. The co-distribution of allysine and collagen is also measured, showing newly deposited collagen with active allysine in the periphery of mature scars. The mouse and fish MI models are shown to be very different in the amount of matured collagen cross-links that form in the MI scars.

The manuscript's clear merit is in the development and utilization of fluorescent imaging to study in excellent spatial detail the localization of actively cross-linking collagen in MI scars (Figure 5a-h). The manuscript also convincingly shows that mouse and fish MI models have very different time courses and collagen organizations. However, since I am not an expert in MI, I am not sure how significant the latter claim is. The manuscript is well-written and I had no problems in following the scientific narrative.

I would like to ask the authors to add the following information/clarifications:

1) The corresponding images to Figs. 2g and 2h of the other tested compounds (TMR-HZN, TMR-Pyr, etc.). To my understanding these images are available because they were used to calculate the values of Fig. 2i. The additional images can be put in the SI, but in my opinion they are critical in convincing the reader of the advantages of TMR-O in practical imaging scenarios.

2) What is the phantom used as the baseline in Fig. 2i? It is not explained in the methods.

3) Please state that the curves in Fig. 2b,c are of the sample that *passed through* the filter. Upon first read of the text, I was confused that it was of the BSA fraction in the filter.

4) I did not understand what the 4th cluster in Fig. 5e-g means. Perhaps a more elaborate explanation of the clustering can be added in the methods.

Point by point response to reviewer comments.

Reviewer #1

Overall, the experimental design is well done and innovative, providing a new method to measure collagen oxidation and cross-link formation in hearts in vivo. It would be much improved if the authors provide any mechanistic insights on mature cross-links between zebrafish and mice. With that said, here are my suggestions and concerns:

We thank the reviewer for their positive response to our paper and for finding our work innovative. We agree that additional mechanistic insights into the differences we have shown between the mature collagen cross-links (pyridinoline and deoxypyridinoline) in the zebrafish and murine hearts would be very valuable. In response to this excellent suggestion, we have performed additional HPLC experiments to assess hydroxylysine content in infarcted murine and zebrafish hearts. A new figure (Figure 7) showing the results of these experiments has been added to the manuscript.

The hydroxylation of lysine on collagen occurs before its oxidation and is required for the formation of both pyridinoline and deoxypyridinoline cross links. (Pyridinoline is also known in its non-abbreviated form as hydroxylsypyrindinoline). We hypothesized, therefore, that hydroxylysine content in newly deposited collagen in zebrafish infarcts would be significantly lower than in murine infarcts. The results of the HPLC analysis shown in Figure 7 confirm our hypothesis. Hydroxyproline content increased significantly between 1-3 weeks after infarction in both the murine and zebrafish hearts, consistent with the deposition of new collagen. Hydroxylysine content in the murine hearts showed a corresponding increase. However, in the zebrafish hearts no increase in hydroxylysine content was seen between 1-3 weeks after infarction. These new data establish the molecular/biophysical mechanism for the differences in pyridinoline and deoxypyridinoline content between the murine and zebrafish hearts: **The lack of hydroxylysine (lysine hydroxylation) in newly deposited collagen in the zebrafish heart impedes its ability to form mature degradation-resistant CCLs (pyridinoline and deoxypyridinoline), which require hydroxylysine for their formation.**

Full details of these new data can be found in Figure 7, supplemental figures S6 and S7, the last 3 paragraphs of the results section, and paragraph 4 of the discussion.

1. In Figure 1, the structure and the on-rates for reaction of TMR-O and TMR-LHZ were similar, why the extent of product formation is different?

We thank the reviewer for raising this point. The key difference between the 2 compounds lies in their off-rates due to very different rates of hydrolysis. We appreciate the opportunity to clarify this and have added the following text to the discussion (paragraph 3):

“The clearest evidence of the dominating effect of hydrolysis is observed in comparing TMR-O and TMR-LHZ which have near identical structures and near identical on-rate reactivity. However, with TMR-O, the slow hydrolysis of the oxime bond results in a much higher accumulation on target tissue in vivo.³⁸ The slow hydrolysis of oximes compared to hydrazones is a known phenomenon resulting from the electron withdrawing nature of the oxygen in oximes compared to nitrogen in hydrazones.^{39,40} This so called “alpha nucleophile” dictates the rate of hydrolysis, as we have recapitulated in our study.”

2. In Figure 3, the signal of TMR-O peaked at 21 days post injury, which means collagen

oxidation and crosslinking peaked at this time point. Why the collagen volume didn't change from 21 to 35 days post injury?

The reviewer raises an interesting point. Although there was a very slight increase in collagen volume at 35 dpi, this was not statistically significant and reflects the regenerative capacity of the zebrafish heart. Collagen volume does not increase in the injured zebrafish heart from day 21- day 35 (despite TMR-O accumulation peaking at day 21) because the collagen in the injury area is also being simultaneously broken down as the heart begins to regenerate.

3. In Figure 5, the study needs more numbers of mouse and zebrafish hearts, and the current data may limit the generalizability of the findings.

We have added additional mice and there are now n=5 mice at all time points. Likewise, for zebrafish there are n=5 zebrafish for each time point. Figure 5 has been updated accordingly.

4. The study primarily present a method on detecting the dynamics of collagen oxidation and cross-links, but have no any mechanistic insights into the underlying processes, such as related gene expression and function on the context of collagen maturation.

Please see our response above regarding the mechanism of mature collagen cross link (CCL) formation. As shown in the newly added Figure 7, the lack of mature CCLs in injured zebrafish hearts reflects the lack of hydroxylysine on newly deposited collagen, which is necessary for the formation of both pyridinoline and deoxypyridinoline.

5. The products of pyridinoline and deoxypyridinoline can be detected even without the aid of TRM-O. Are there any specific small molecules that can be used to remove pyridinoline and deoxypyridinoline for treating CCL?

The reviewer raises an interesting question. To the best of our knowledge there are no small molecules that can result in the removal of pyridinoline and deoxypyridinoline cross-links once they have formed. It will be interesting in future studies to determine whether any small molecules can prevent the formation of pyridinoline and deoxypyridinoline in the mammalian heart, replicating the natural history of CCLs in the zebrafish heart.

6. In line 136, TMR-HDZ should be TMR-HZD?

Changed, thank you

Reviewer #2 (Remarks to the Author):

The study is original and timely and the comparison between two different models adds considerable strength. The findings will be of interest to a broad range of investigators working in regenerative medicine and matrix biology. The studies appear well done and the results are highly interesting, particularly that early cross linking appears very similar between the two models and the fact that adult mouse collagen crosslinking appears dynamic over a longer period of time that previously thought.

We thank the reviewer for their positive response to our manuscript.

1 – On line 110 the authors state that the butyraldehyde, used as a target aldehyde for the in vitro investigations in Figure 1, has a “similar structure to the in vivo aldehyde target allysine.” How similar are they? How applicable is this data to the in vivo findings? I appreciate that the TMR-O probe performs best in all investigations but can more be learnt from investigating the other probes in vivo considering their different observed on and off rates?

We appreciate the reviewer’s point and have added a figure to the supplement (figure S1) with the structures of butyraldehyde and allysine, which shows that the moiety targeted by the probe (i.e. the aldehyde) is essentially identical between the two structures.

The reviewer also makes a good point regarding replicating the in vitro findings of probe affinity in vivo. We chose to do this by investigating the in vivo binding of the library of probes to healthy zebrafish bulbous, which contains large amounts of allysine. These data are summarized in the graph shown in Figure 2i, where the signal from TMR-O was significantly higher than all the other probes. To complement the data shown in Figure 2i, we have added a new figure to the supplement (Figure S4) showing images of the various probes binding to the zebrafish bulbous. The signal from TMR-O in the bulbous is markedly higher than the other probes. These in vivo data faithfully recapitulate the in vitro data and show that TMR-O performs best in both settings.

We very much agree with the reviewer that much can be learned about probe performance from their on and off rates. We have found the off rates to be particularly important. For instance, when comparing TMR-O to the linear hydrazine-based probe TMR-LHZ, we observe that although these two probes have near identical kinetics for binding (i.e. on-rates), TMR-O performs much better in vivo because its off-rate due to hydrolysis is far slower than the off-rate (hydrolysis) of TMR-LHZ. This is indicated by the 2-fold higher fluorescence of the bulbous of fish injected with TMR-O compared to TMR-LHZ (Figure 2i and Figure S4)

2 – Can the authors comment on why different delivery methods were chosen for the mice (intravenous) and the zebrafish (intraperitoneal). Could these different methods result in different accumulation kinetics?

Intravenous (IV) injections into the tail vein in mice produced very consistent probe kinetics. However, in zebrafish IV injections are done into the retro-orbital venous plexus and did not result in consistent and reliable probe kinetics. In contrast, we found that probe kinetics in zebrafish were very consistent and reliable after intraperitoneal (IP) injection.

We agree with the reviewer that the peak concentration of probe in the blood will occur earlier with IV than IP injection, but over time both routes will result in similar concentrations and amounts of probe in the blood. However, the time chosen for imaging was not based on peak/early concentrations but rather on the time needed for complete clearance of non-bound probe from the circulation, thereby minimizing any non-specific background signal. We determined this clearance time by injecting non-targeted tetramethylrhodamine (TMR), the base fluorophore used in all our probes, into zebrafish (IP) and mice (IV). TMR was completely cleared from the murine circulation by 1.5 hour post IV injection. In zebrafish, since absorption into the blood after IP injection is more gradual, it took 4 hours for TMR to be absorbed and then eliminated from the circulation in zebrafish. The 1.5-hour and 4-hour time points for imaging were based on these elimination times. We have now added an explanation to clarify why these delivery routes and imaging times were chosen to the methods section.

3 – Following on from this the authors describe different labelling timelines for mice and fish (90 mins vs 4 hours, respectively – data in Figures 3 and 4) but on line 314 say that the labelling in mice was “substantially higher than in the zebrafish.” Can these data really be compared? I think a direct quantitative comparison would be needed to make this conclusion.

We agree with the reviewer that the fluorescent signal from the infarct/injury can be influenced by small differences in the amount of probe injected, probe kinetics and background signal. To address this, we measured target-background ratios (ratio of fluorescence in the injury to the healthy/remote region of the heart) in all cases. Normalizing the data in this way accounts for any variations in kinetics within and between groups. We do agree with the reviewer, however, that directly measuring collagen content in the infarct by HPLC is a more definitive method than probe target-to-background ratio. As shown in Figure 7, we have now quantified collagen (hydroxyproline) content in the infarcts by HPLC at week 1 and 3. The hydroxyproline peaks (collagen content) in mice are substantially higher than in zebrafish consistent with the higher uptake of the TMRO-probe in mice.

4 – On line 396 the authors state “The maturation of collagen cross-links occurs by non-enzymatic processes and is considered to be spontaneous (Fig. 6b -c), but can be influenced by the exact peptide-sequence and glycosylation state of collagen.²⁸” Have the authors investigated differences in the peptide sequence and glycosylation state of collagens between mice and zf as a way to explain their findings?

This is an excellent point that we have certainly considered. However, since all mature collagen-cross links (pyridinoline and deoxypyridinoline) require hydroxylysine for their formation, we have chosen to focus for now on differences in hydroxylysine (lysine hydroxylation) content between zebrafish and mice. Figure 7 describes these new data. Using HPLC we found that hydroxylysine content in newly deposited collagen in injured zebrafish hearts was significantly lower than in mice. The lack of hydroxylysine in newly deposited collagen in the zebrafish heart limits its ability to form pyridinoline and deoxypyridinoline cross-links, for which hydroxylysine is required. It is certainly possible that differences in glycosylation and the exact peptide sequence of zebrafish collagen could also be playing a role, and we will investigate this in future studies. However, our recent data (Figure 7) suggest that differences in lysine hydroxylation play a key role in the absence of mature collagen cross-links in the injured zebrafish heart.

5 – On line 409 and in the legend of Figure 6 the authors say that there are “similar” levels of hydroxyproline in the zf and mouse samples, is there a way to show this data? Or for these levels to be more precise? This would aid in the robustness of this data.

We appreciate this opportunity to clarify this and, at the reviewer’s suggestion, have also added a new figure to the supplement (figure S5) to depict our approach graphically. The zebrafish heart is ~1% the size of the mouse heart and amount of tissue it provides for analysis is much lower. To ensure that HPLC for collagen cross-links was performed on samples containing similar amounts of collagen, the amount of hydroxyproline content in the final samples was standardized. As shown in Figure S5, the amount of hydroxyproline in mouse samples is ~100-fold higher than in the zebrafish samples. To account for this difference, we used 100-fold more concentrated zebrafish tissue digest to create mouse and zebrafish samples with equivalent concentrations of hydroxyproline. The HPLC traces of mature CCLs (pyridinoline and

deoxypyridinoline) shown in Figure 6 were thus all acquired in samples with similar concentrations of collagen (hydroxyproline).

Minor Points for clarification:

Figure 5 legend, graphs in f and g – is each point one mouse? The legend says “did not change significantly over time”. How were the statistical tests done?

We thank the reviewer for pointing this out. Yes, each point refers to one mouse. We performed one-way ANOVA with Tukey’s post-hoc analysis. We have now included this in the Figure legend.

Line 391 – “resides” should be “residues”?

Corrected, thank you.

Line 392 – insert “other” after “each”

Added, thank you.

Reviewer #3 (Remarks to the Author):

The manuscript's clear merit is in the development and utilization of fluorescent imaging to study in excellent spatial detail the localization of actively cross-linking collagen in MI scars (Figure 5a-h). The manuscript also convincingly shows that mouse and fish MI models have very different time courses and collagen organizations. However, since I am not an expert in MI, I am not sure how significant the latter claim is. The manuscript is well-written and I had no problems in following the scientific narrative.

We thank the reviewer for their positive response to our work.

I would like to ask the authors to add the following information/clarifications:

1) The corresponding images to Figs. 2g and 2h of the other tested compounds (TMR-HZN, TMR-Pyr, etc.). To my understanding these images are available because they were used to calculate the values of Fig. 2i. The additional images can be put in the SI, but in my opinion they are critical in convincing the reader of the advantages of TMR-O in practical imaging scenarios.

We thank the reviewer for this suggestion; these images are indeed available and we have now included them in a new figure in the supplement (figure S4).

2) What is the phantom used as the baseline in Fig. 2i? It is not explained in the methods.

We apologize for not explaining this fully. To account for potential concentration variations in the formulated dose injected into each zebrafish, a calibration phantom using the same formulation of the probe was imaged with the bulbous in each case. The signal in each bulbous was normalized by the signal in its corresponding calibration phantom. We have now included this information in the methods section.

3) Please state that the curves in Fig. 2b,c are of the sample that *passed through* the filter. Upon first read of the text, I was confused that it was of the BSA fraction in the filter.

The figure caption has been changed to clarify this, thank you.

4) I did not understand what the 4th cluster in Fig. 5e-g means. Perhaps a more elaborate explanation of the clustering can be added in the methods.

We thank the reviewer for this suggestion. We have added the following text to the methods section under "Image Analysis"

For k-means cluster analysis, a region of interest (ROI) was defined to include the entire visible area of injury. Segmentation of the TMR-O signal in this ROI was performed using k-means clustering (ImageJ) based on the brightness of the TMR-O signal. The fourth (brightest) cluster was presumed to contain pixels with the highest concentration of freshly oxidized (and therefore freshly deposited) collagen. The percentage of the overall ROI occupied by pixels in the fourth cluster and the average intensity of this cluster were then determined.

REVIEWERS' COMMENTS

Reviewer #1 (Remarks to the Author):

The authors have carefully addressed my comments, and I have no further suggestions.

Reviewer #2 (Remarks to the Author):

All my comments have been addressed. The inclusion of new mechanistic findings is welcome and further strengthens the work already described.

Reviewer #3 (Remarks to the Author):

The authors have addressed all my concerns in the revised manuscript.